# ON THE GEOMETRY OF UNCERTAINTY: EQUIVARIANT UNCERTAINTY ESTIMATION FOR MOLECULAR VECTOR PROPERTIES

## ABSTRACT

Accurate and reliable uncertainty estimation (UE) for vector-valued physical properties is crucial for scientific discovery in fields like drug and materials discovery. For example, atomic forces are central for finding optimal structures and identifying equilibrium systems, and a fundamental requirement of these vectors is that they must be equivariant to 3D rotations. However, existing UQ methods often fail to respect these geometric constraints, leading to poorly calibrated uncertainty and degraded predictive performance. To address the problem, we introduce a novel framework for equivariant multivariate evidential regression. Our key contribution is a new parameterization for the covariance matrix, which is provably equivariant, identifiable, and correctly parameterized. We further introduce a calibration metric based on the Mahalanobis distance to rigorously assess equivariant multivariate uncertainty. Extensive experiments on molecular property prediction benchmarks, including MD17, OC20, and QM7-X, show that our framework consistently outperforms established baselines, achieving state-of-the-art predictive accuracy with well-calibrated uncertainty. This work provides a principled and practical approach to uncertainty estimation for equivariant vector-valued properties, paving the way for more trustworthy machine learning applications in scientific discovery.

## 1 INTRODUCTION

The accurate prediction of molecular or crystal properties using their 3D structures, is fundamental to modern computational science, enabling applications from drug discovery to materials design. The gold standard for these predictions, quantum mechanical (QM) calculations (Hohenberg & Kohn, 1964; Kohn & Sham, 1965), remains prohibitively expensive for large-scale simulations. To overcome this bottleneck, learning-based approaches, particularly Graph Neural Networks (GNNs), have emerged as powerful surrogates, capable of learning complex atomic interactions from QM data and accelerating simulations by orders of magnitude without sacrificing the solution quality. (Schütt et al., 2017b;a; Gilmer et al., 2017; Klicpera et al., 2020; Choi et al., 2025)

Accurately predicting the properties of molecules and crystals is a crucial early step in materials and drug discovery. These predictions are typically followed by costly and time-consuming experimental validation of the most promising candidates identified through learning-based approaches. In this context, it has become essential to reliably quantify the uncertainty of such predictions, ensuring that resources are allocated efficiently and that confidence in the models is strengthened (Svensson et al., 2025; Zhong et al., 2020; Abed et al., 2024). Beyond uncertainty quantification, it is equally important that predictive models respect the fundamental symmetries of molecular and crystalline systems. Equivariance (Thomas et al., 2018; Batzner et al., 2022; Batatia et al., 2022; Zitnick et al., 2022; Passaro & Zitnick, 2023) ensures that predictions remain consistent under rotations, translations, or permutations, thereby aligning the model with the underlying physics.

In a nutshell, current molecular/crystal property prediction systems mainly cover two categories of quantities: (1) scalar quantities such as total energy or log partition coefficients ($\log P$); and (2) vectorized quantities such as atomic force. While scalar quantities can be used to describe several important properties of chemicals, vectorized quantities often provide a more detailed and

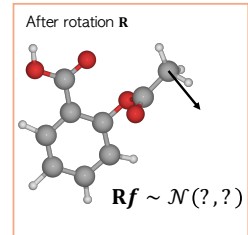 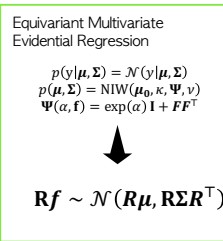

Figure 1: An illustration of equivariant uncertainty for a vector property $\boldsymbol{f}$, such as an atomic force. Before rotation, $\boldsymbol{f}$ following a multivariate normal distribution. After rotation, the $\boldsymbol{f}$ should follow a specific multivariate normal distribution as to reflect the equivariance. Our proposed method, Equivariant Multivariate Evidential Regression, is designed to directly learn this rotation-aware uncertainty.

fine-grained characterization of molecular interactions. For example, accurate force predictions are essential for tasks such as geometry optimization, Molecular Dynamics (MD) simulations, and understanding reaction pathways. Moreover, reliably estimating the uncertainty in these vectorized predictions is particularly important, since even small errors in predicted forces can accumulate and lead to qualitatively incorrect trajectories in long-timescale simulations. By quantifying the uncertainty of vectorized outputs, researchers can better assess the stability of simulated systems, identify regions of chemical space where models are less reliable, and guide experimental or high-fidelity computational validation more effectively (Li et al., 2022; Stocker et al., 2022).

In current uncertainty estimation (UE), particularly deep learning–based approaches (Amini et al., 2020; Soleimany et al., 2021; Wenzel et al., 2020; van Amersfoort et al., 2022), neural networks are trained to output not a single prediction but a parameterized distribution over the quantities of interest. For scalar quantities, these approaches naturally satisfy equivariance constraints, as the distribution remains invariant under transformations. However, for vectorized quantities, existing methods often violate equivariance: the predicted distribution changes inconsistently when rotations are applied to the input. Most of the current UE techniques were originally designed for scalar outputs and can be broadly divided into two families. Sampling-based methods, such as ensembles and dropout (Gal & Ghahramani, 2016; Wen et al., 2020), approximate a predictive distribution by generating and aggregating multiple predictions. In contrast, sampling-free approaches, including evidential regression (Amini et al., 2020; Kopetzki et al., 2021) and stochastic variational Gaussian processes (Rasmussen & Williams, 2006; Hensman et al., 2013; 2015; Wollschläger et al., 2023), directly output the parameters of a distribution in a single forward pass, often with improved computational efficiency.

While sampling-based approaches can, in principle, be extended to respect equivariant constraints, such extensions are computationally expensive due to the need for large numbers of samples and symmetry operations. Sampling-free approaches, on the other hand, can be generalized from univariate to multivariate distributions by replacing scalar predictive distributions with their multivariate counterparts. However, a naïve extension such as Soleimany et al. (2021) generally breaks equivariance, since the resulting distribution fails to transform consistently with the underlying physical symmetries. This highlights the need for new frameworks that can simultaneously achieve computational tractability and preserve equivariance in the UE of vectorized quantities.

To bridge this critical gap, we introduce a novel framework for Equivariant Multivariate Evidential Regression (EMER). Our core contribution is a new, provably equivariant parameterization for the covariance matrix that allows a model to produce reliable and geometrically consistent UE for vector outputs. Furthermore, we propose a robust multivariate calibration metric based on the Mahalanobis distance to properly evaluate the quality of vector-based uncertainty. We demonstrate on molecular force and dipole moment prediction benchmarks that our method not only produces well-calibrated uncertainty but also achieves state-of-the-art predictive accuracy, outperforming existing UE baselines.

To summarize, our contributions are:

- We first formalize the problem of equivariant UE for vector-valued physical properties, establishing the theoretical requirements for geometrically consistent uncertainty.

- We propose a novel equivariant multivariate evidential regression framework featuring a new covariance parameterization that is provably equivariant under $SO(3)$ rotations.

- We introduce a Mahalanobis distance-based calibration score for equivariant multivariate uncertainty estimates, overcoming the inapplicability of standard scalar metrics in this setting.

- We empirically demonstrate that our method sets a state of the art accuracy and calibration compared to previous approaches applicable to the equivariant case..

## 2 RELATED WORK

### 2.1 GRAPH NEURAL NETWORK

GNNs have emerged as a powerful framework for molecular property prediction (Schütt et al., 2017a;b; Gasteiger et al., 2021). The core of most GNNs is the message-passing paradigm, where node representations are iteratively updated by aggregating information from their local neighborhoods. (Gilmer et al., 2017) However, standard message-passing architectures often fail to capture the inherent geometric symmetries. For instance, in molecular modeling, some properties (e.g. force) are dependent of its orientation in 3D space. A standard GNN, which might only use interatomic distances, discards crucial directional information.

To address this limitation, equivariant GNNs (EGNN) were developed to explicitly incorporate geometric symmetries, particularly the rotational symmetries of 3D space ($SO(3)$) symmetry). (Thomas et al., 2018; Batzner et al., 2022; Batatia et al., 2022; Zitnick et al., 2022; Passaro & Zitnick, 2023) The principle of equivariance ensures that as the input coordinates are rotated, the intermediate feature vectors within the network transform in a corresponding, predictable manner. Mathematically, a function $f$ (like a neural network layer) is equivariant to a rotation $\mathcal{R} \in SO(3)$ if rotating the input coordinates $\mathbf{x}$ produces an identically rotated output:

$$f(\mathcal{R}\mathbf{x}) = \mathcal{R}f(\mathbf{x})$$

This embeds a strong inductive bias for 3D geometry directly into the model's architecture. By preserving this geometric structure, equivariant models can predict both invariant scalar properties (e.g., energy) and equivariant vector properties (e.g., forces, dipoles), leading to significantly improved data efficiency and generalization performance.

### 2.2 UNCERTAINTY ESTIMATION

In the area of machine learning potentials, there is an urgent need for calibrated UE methods, as a high-error prediction during a MD simulation would lead to wrong conclusions. While numerous UE strategies exist, several have become common in molecular modeling due to their differentiability and applicability to energy predictions.

**Ensembles**, often regarded as a gold standard for UE, are predicated on training a diverse set of models on identical data (Kuleshov et al., 2018b). The prediction variance across the ensemble serves as a uncertainty metric. Despite their high accuracy, ensembles impose a significant computational burden, as the cost of both training and inference scales linearly with the number of models, creating a substantial bottleneck.

**Monte Carlo (MC) Dropout** offers a more computationally efficient approximation of Bayesian inference in deep neural networks (Gal & Ghahramani, 2016). This technique involves performing multiple stochastic forward passes at inference time with dropout layers enabled. Depending on how dropout were performed, it can be categorized into Bernoulli Dropout (Gal & Ghahramani, 2016), Gaussian Dropout (Wang & Manning, 2013) and Variational Dropout (Kingma et al., 2015; Liu et al., 2019). Each pass samples a different sub-network, and the variance of the resulting predictions is used to estimate model uncertainty. While this approach eliminates the need to train multiple models, the requirement for repeated forward passes can still be prohibitive.

**Evidential Regression** is a sampling-free UE method that learns the parameters of a higher-order evidential distribution (e.g., a Normal-Inverse-Gamma distribution) from which the predictive uncertainty can be derived analytically in a single forward pass (Amini et al., 2020; Soleimany et al.,

2021). However, this method is formulated for independent scalar outputs. When naively applied to a vector by treating each component separately, the resulting uncertainty estimate fails to capture the covariance between components, thereby violating the crucial property of rotational equivariance.

**Stochastic Variational Gaussian Process (SVGP)** is a scalable Bayesian method that provides calibrated UE for molecular property prediction (Hensman et al., 2015). By using a sparse variational framework with inducing points, SVGP avoids the cubic complexity of exact Gaussian Processes (Rasmussen & Williams, 2006) and can be trained efficiently on large molecular datasets with mini-batch optimization. SVGP is often combined with Deep Kernel Learning (DKL) (Wilson et al., 2016), where a deep neural network maps molecular structures into a feature space for the SVGP. Building on this, Wollschläger et al. (2023) introduced the Localizing Neural Kernel (LNK), which fits an SVGP to each atom's embedding and sums the per-atom contributions, consistent with the principle that total energy is additive over atoms. However, the nonlinear mappings used in LNK generally break geometric symmetries, meaning that equivariance is not preserved for vector properties.

**Calibration Score** To ensure the trustworthiness of a UE method, the quality of its uncertainty estimates must be evaluated. A primary means of this assessment is the calibration score (Kuleshov et al., 2018a). A model is considered well-calibrated if its predicted uncertainty accurately reflects its true prediction error. This evaluation is performed by comparing the model's predicted Cumulative Distribution Function (CDF) against the empirically observed distribution. The discrepancy between the expected confidence and the actual observed frequency is often visualized in a reliability diagram, and this gap is quantified by a single metric such as the calibration score (Kuleshov et al., 2018a) or miscalibration area (Tran et al., 2020).

## 3 PROPOSED METHODS

In this section, we first give a brief introduction to the equivariant learning approaches and previous UE approaches. We then demonstrate that conventional UE methods fail to preserve rotational equivariance, a property essential for tasks in the physical systems. To resolve this incompatibility, we introduce a novel framework that provides robust uncertainty estimates while rigorously maintaining geometric symmetries.

### 3.1 PRELIMINARIES

**Equivariant Predictor and Uncertain Estimation** Given a learning-based approach $\hat{y} = f(x|\Theta)$, where $x$ represents the input data and $\hat{y}$ is the predicted vector property, equivariant methods provide a prediction $\hat{y}$ that rotates consistently with the input $x$, satisfying $\mathcal{R}(\hat{y}) = f(\mathcal{R}(x); \Theta)$ for any rotation operator $\mathcal{R}$. In UE, rather than providing a single deterministic output, the goal is to quantify the prediction uncertainty. Therefore, instead of simply a point estimate of $\hat{y}$, the model must estimate a distribution. Particularly for an equivariant learning approach, this distribution must itself be equivariant. This imposes the constraint that for any rotation operator $\mathcal{R}$, the distribution must satisfy $p(\mathcal{R}(y)|\mathcal{R}(x); \Theta) = p(y|x; \Theta)$.

**Multivariate Evidential Regression** To address the need for an efficient, single-pass and equivariant UE method for vectors, we turn to Multivariate Evidential Regression (MER). As an extension of evidential regression for scalar targets (Amini et al., 2020; Soleimany et al., 2021), MER was introduced by Meinert & Lavin (2021) to handle multivariate outputs. The core innovation is to replace the Normal-Inverse-Gamma prior with a Normal-Inverse-Wishart (NIW) distribution, as

$$p(y) = \mathcal{N}(\mu, \Sigma) \qquad (\mu, \Sigma) \sim \text{NIW}(\mu_0, \kappa, \Psi, \nu) \tag{1}$$

Here, $\mu_0 \in \mathbb{R}^n$ is the location vector representing the estimated mean, with $n$ denotes the dimension of the target vector ($n = 3$ for 3D forces). $\kappa \in \mathbb{R}_{>0}$ is a scalar scaling factor reflecting the confidence in the mean estimation. $\Psi \in \mathbb{S}_{++}^n$ is an $n \times n$ symmetric positive-definite (SPD) scale matrix, and $\nu \in \mathbb{R}$ represents the degrees of freedom, which must satisfy $\nu > n + 1$ to ensure a valid mean and variance.

A neural network $f_\theta$ learns a mapping from an input graph $x$ to the four NIW parameters, i.e., $(\mu_0, \kappa, \Psi, \nu) = f_\theta(x)$. Following the setting used in Meinert & Lavin (2021) that $\kappa = r\nu$ and

$r = 1$, the loss function can be write as

$$L_{\text{NLL}_i} = -\log p(\boldsymbol{y}_i \mid \boldsymbol{\mu}_0, \kappa, \boldsymbol{\Psi}, \nu) = \log \Gamma\left(\frac{\nu - n + 1}{2}\right) - \log \Gamma\left(\frac{\nu + 1}{2}\right)$$

$$+ \frac{d}{2}\log(1 + \nu) - \frac{\nu}{2}\log|\boldsymbol{\Psi}| + \frac{\nu + 1}{2}\log\left|\boldsymbol{\Psi} + \frac{1}{1 + \mu}(\boldsymbol{y}_i - \boldsymbol{\mu}_0)(\boldsymbol{y}_i - \boldsymbol{\mu}_0)^\top\right| \tag{2}$$

where $\Gamma(\cdot)$ denotes the Gamma function.

The estimation of predictions and uncertainties are

$$\hat{\boldsymbol{y}} = \mathbb{E}[\boldsymbol{\mu}] = \boldsymbol{\mu}_0 \qquad \mathbb{E}[\boldsymbol{\Sigma}] = \frac{\boldsymbol{\Psi}}{\nu - d - 1} \tag{3}$$

To ensure the matrix $\boldsymbol{\Psi}$ is symmetric and Positive Definiteness (PD), MER construct it from its Cholesky factor $\mathbf{L}$, which is parameterized by the network outputs. That is, instead of predicting $\boldsymbol{\Psi}$ directly, the network yields the elements of the lower-triangular matrix $\mathbf{L}$. For the $i$-th data point, the components of $\mathbf{L}_i$ are defined as:

$$(L_i)_{jk} = \begin{cases} \exp\left(\boldsymbol{l}_j^{(i)}\right) & \text{if } j = k, \\ \boldsymbol{l}_{jk}^{(i)} & \text{if } j > k, \\ 0 & \text{otherwise.} \end{cases} \tag{4}$$

The matrix is then assembled via $\boldsymbol{\Psi}_i = \mathbf{L}_i \mathbf{L}_i^\top$.

However, directly applying the MER framework to equivariant UE is ill-posed. In MER, a network outputs the scalar components of a lower-triangular matrix $\mathbf{L}$. An equivariant network producing scalar outputs will necessarily yield rotationally invariant scalars, making the resulting matrices $\mathbf{L}$ and $\boldsymbol{\Psi} = \mathbf{L}\mathbf{L}^\top$ also invariant to input rotations, meanwhile in an equivariant system $\boldsymbol{\Psi}$ should rotate in an ideitical way as in Equation (5)[1]. Alternatively, constructing $\mathbf{L}$ from an equivariant vector output is challenging, since the use of nonlinear exponentiation function for diagonal elements breaks the equivariance. In equivariant models like EGNN, all non-linear functions must be specially designed to preserve equivariance, such as norm-based (Geiger & Smidt, 2022) and gated activations (Weiler et al., 2018a;b). This requirement for specialized nonlinearities also explains why other UE methods, such as SVGP, cannot be directly applied to vector-valued outputs.

## 3.2 Equivariant Multivariate Evidential Regression

While MER and other single-pass UE methods struggle to directly predict a valid equivariant covariance matrix, sampling-based approaches like ensembles or MC dropout natively satisfy this. In these cases, the covariance is estimated empirically from a set of $K$ predictions $\{\mathbf{v}^{(k)}\}$ as $\hat{\boldsymbol{\Sigma}} = \frac{1}{K-1}\sum_{k=1}^{K}(\mathbf{v}^{(k)} - \bar{\mathbf{v}})(\mathbf{v}^{(k)} - \bar{\mathbf{v}})^\top$, where $\bar{\mathbf{v}}$ is the sample mean. This empirical covariance is guaranteed to be Positive Semidefinite (PSD) and to satisfy the equivariance of

$$\boldsymbol{\Sigma}(\boldsymbol{R}\boldsymbol{x}) = \boldsymbol{R}\boldsymbol{\Sigma}(\boldsymbol{x})\boldsymbol{R}^\top. \tag{5}$$

Now we focus on how to make MER equivariant. Naive equivariant alternatives, such as parameterizing the covariance via an outer product of a equivariant matrix ($\boldsymbol{\Sigma} = \boldsymbol{F}\boldsymbol{F}^\top$) satisfy this, but it create an ill-posed optimization problem due to overparameterization. A $n$-dimensional symmetric PSD matrix $\boldsymbol{\Sigma}$ has $M = n(n+1)/2$ unique, independent parameters. The general factor matrix $\boldsymbol{F}$, however, has $N = n^2$ parameters. For a 3D case (e.g., molecular forces, where $n = 3$), this results in $N - M = 3$ redundant parameters, making $\boldsymbol{F}$ overparameterized. Besides, the solution for $\boldsymbol{F}$ is also not unique. Let $\boldsymbol{Q}$ be any $n \times n$ orthogonal matrix, satisfying $\boldsymbol{Q}\boldsymbol{Q}^\top = \mathbf{I}$. If $\boldsymbol{F}$ is a factor that generates $\boldsymbol{\Sigma}$, then the rotated factor $\boldsymbol{F}' = \boldsymbol{F}\boldsymbol{Q}$ generates an identical covariance matrix:

$$\boldsymbol{\Sigma}' = \boldsymbol{F}'(\boldsymbol{F}')^\top = (\boldsymbol{F}\boldsymbol{Q})(\boldsymbol{F}\boldsymbol{Q})^\top = \boldsymbol{F}\boldsymbol{Q}\boldsymbol{Q}^\top\boldsymbol{F}^\top = \boldsymbol{F}\mathbf{I}\boldsymbol{F}^\top = \boldsymbol{\Sigma}$$

Since the set of orthogonal matrices $\boldsymbol{Q}$ is infinite, the loss function $L(\boldsymbol{\Sigma}) = L(\boldsymbol{F}\boldsymbol{F}^\top)$ possesses an infinite, continuous manifold of equivalent minima with respect to the predicted factor $\boldsymbol{F}$.

---

[1]By Equation (3) there is only a scale factor between $\Psi$ and the expectation of $\Sigma$.

To resolve this, we propose Equivariant Multivariate Evidential Regression, which introduces a novel parameterization for the covariance structure that is inherently equivariant by construction.

**Proposition 3.1** *Let the scale matrix $\boldsymbol{\Psi}$ in the Normal-Inverse-Wishart (NIW) distribution be parameterized as*

$$\boldsymbol{\Psi}(\alpha, \boldsymbol{F}) = \exp(\alpha)\mathbf{I} + \boldsymbol{F}\boldsymbol{F}^{\top}$$

*where $\alpha \in \mathbb{R}$ is an invariant scalar and $\boldsymbol{F} \in \mathbb{R}^{3 \times k}$ ($k < 3$) is an equivariant matrix, satisfies the following properties:*

*(i)* $\boldsymbol{\Psi}$ *is guaranteed to be Positive Definiteness (PD), as $\exp(\alpha) > 0$.*

*(ii)* *The matrix $\boldsymbol{\Psi}$ is equivariant: $\boldsymbol{\Psi}' = \mathbf{R}\boldsymbol{\Psi}\mathbf{R}^{\top}$ for a rotation $\mathbf{R}$.*

*(iii)* *The parameters $(\alpha, \boldsymbol{F})$ are unique from $\boldsymbol{\Psi}$ up to the sign of $\boldsymbol{F}$, i.e., $\boldsymbol{\Psi}(\alpha, \boldsymbol{F}) = \boldsymbol{\Psi}(\alpha', \boldsymbol{F}') \implies \alpha = \alpha'$ and $\boldsymbol{F}' = \pm\boldsymbol{F}$.*

**Proof Sketch** The proposed parameterization $\boldsymbol{\Psi} = \exp(\alpha)\mathbf{I} + \boldsymbol{F}\boldsymbol{F}^{\top}$ is guaranteed to be PD because it is the sum of a PD isotropic term ($\exp(\alpha)\mathbf{I}$) and a PSD low-rank term ($\boldsymbol{F}\boldsymbol{F}^{\top}$). Equivariance is inherently satisfied because the invariant scalar $\alpha$ is associated with $\mathbf{I}$, and the equivariant matrix $\boldsymbol{F}$ is used only in the outer product, ensuring $\boldsymbol{\Psi}$ transforms correctly under rotation ($\boldsymbol{\Psi}' = \mathbf{R}\boldsymbol{\Psi}\mathbf{R}^{\top}$). Identifiability is established via the eigenvalue decomposition of $\boldsymbol{\Psi}$. The scalar $\exp(\alpha)$ is uniquely determined by the two smallest (degenerate) eigenvalues. The magnitude of $\boldsymbol{F}$ is then determined by the remaining principal eigenvalue, and its direction by the unique principal eigenvector, resulting in uniqueness up to the sign flip of $\boldsymbol{F}$ ($\boldsymbol{F}' = \pm\boldsymbol{F}$).

**Avoiding pitfalls** To enhance the stability of our UE and address the challenges of combining a large EGNN with the MER and EMER heads, we integrated dropout into them (MER-Dropout and EMER-Dropout). Although EMER utilizes a NIW prior for robust evidential uncertainty, the inherent complexity and capacity of the deep GNN encoder can still risk learning an overly confident or poorly conditioned representation. This phenomenon has been observed previously in DKL and is referred to as feature collapse. (Ober et al., 2021; van Amersfoort et al., 2022) By employing dropout, we effectively regularize the feature space, ensure the predicted covariance matrix remains well-conditioned and the overall uncertainty estimate is well-calibrated and robust across different molecular environments. In comparison to MC Dropout (Gal & Ghahramani, 2016), where dropout is applied during both training and inference, here dropout is used only during the training stage. As a result, it serves solely as a regularization mechanism and does not provide UE. This regularization strategy has also been successfully employed in Wollschläger et al. (2023) and Tran et al. (2019).

The parameterized form of $\boldsymbol{\Psi} = \exp(\alpha)\mathbf{I} + \boldsymbol{F}\boldsymbol{F}^{\top}$ represents a trade-off between model expressiveness and computational efficiency. While sampling-based methods (e.g., Ensembles, MC Dropout) offer a non-parametric covariance estimate capable of capturing arbitrary uncertainty distributions, they incur an inference cost that scales linearly with the number of samples $K$. This is prohibitive for high-throughput applications, such as ab initio MD simulations. EMER, by contrast, achieves its single-pass efficiency while maintaining a physically grounded expressiveness sufficient to capture the necessary rotational equivariance and low-rank anisotropy typical of molecular force fields. This design choice prioritizes tractability and physical symmetry over the full, but computationally costly, expressiveness of non-parametric approaches.

## 3.3 A Principled Calibration Metric for Equivariant Vectors

Evaluating the calibration of UE for equivariant vectors presents a unique challenge. Standard univariate metrics like the probability integral transform are inapplicable, while common multivariate checks that assess marginal distributions are ill-suited for physical vectors due to the equivariance. To address this, we introduce a calibration metric based on the Mahalanobis distance. (Bishop & Nasrabadi, 2006) This distance provides a scalar measure of the error between the true vector $\mathbf{v}_{\text{true}}$ and the predicted mean $\boldsymbol{\mu}$, normalized by the predicted covariance matrix $\boldsymbol{\Sigma}$:

$$d^2 = (\mathbf{v}_{\text{true}} - \boldsymbol{\mu})^{\top}\boldsymbol{\Sigma}^{-1}(\mathbf{v}_{\text{true}} - \boldsymbol{\mu}).$$

This metric is powerful because it is inherently invariant to rotations and accounts for the predicted correlations between vector components, essentially measuring how "surprising" an observation is given the model's predicted uncertainty ellipsoid.

This formulation provides a direct statistical test for calibration. For a perfectly calibrated model, the squared Mahalanobis distances $d^2$ computed over a test set must follow a chi-squared ($\chi_n^2$) distribution with $n$ degrees of freedom, where $n$ is the dimension of the vector, as $d^2 \sim \chi_n^2$. We can therefore assess calibration by comparing the empirical CDF of the observed $d^2$ values against the theoretical CDF of the $\chi_n^2$ distribution. Any deviation from the identity line on this plot indicates miscalibration.

### 3.4 OVERALL FRAMEWORK

Our framework is built upon a standard EGNN, which we adapt for UE. As shown in Figure 3 in the Appendix, in addition to the conventional head that predicts the target vector, our method requires two additional uncertainty heads. These heads outputs the parameters for an evidential distribution: two scalars, $\nu$ and $\alpha$, and an equivariant matrix, $\boldsymbol{F}$. These parameters are then used to construct the NIW scale matrix $\boldsymbol{\Psi}$. The entire network is trained end-to-end by minimizing the NLL of the evidential distribution. To implement MER-Dropout and EMER-Dropout, we integrate dropout layers within the network, ensuring they are active only during the training phase. Finally, the model's predictive accuracy and uncertainty quality are evaluated during testing.

## 4 EXPERIMENTS

We first compare established UE techniques with our proposed approach. We compared exising sampling-based methods including ensembles, MC Dropout (Gal & Ghahramani, 2016), as well as two sampling free methods, Classification And Regression Diffusion (CARD) (Han et al., 2022) and MER that do not satisfy the equivariance. Calibration quality is measured using the Mahalanobis distances-based Calibration Score defined in Section 3. Except for it, we also measure the Raw Calibration Score (RCal) defined in (Wollschläger et al., 2023; Kuleshov et al., 2018a) and well as negative log likelihood (NLL). Since there are 3 dimensions, we only report the maximum RCal and NLL. Additional details about the metrices are provided in the Appendix. Unless otherwise specified, for each backbone network we adopt the hyperparameter settings reported in the original papers.

We evaluate them on two quantum chemistry prediction tasks using established benchmarks. For force prediction, we use two datasets: MD17 (Chmiela et al., 2017), which contains energies and atomic forces from MD simulations of small molecules, and Open Catalyst 2020 (OC20) (Chanussot et al., 2021), which targets heterogeneous catalysis and provides adsorption energies and forces for diverse catalyst–adsorbate pairs. For OC20, we randomly select 100k samples from each dataset for training due to computational resource limitations. For dipole-moment prediction, we use QM7-X (Hoja et al., 2021), which provides full vector dipole moments rather than only their magnitudes. Force models are trained with a combined energy-and-force loss, whereas dipole-moment models are trained with a single loss.

For MD17, the training of forces follows the standard energy-conserving protocol: forces are obtained as the negative gradient of the energy (Klicpera et al., 2020), and models are trained by minimising a combined energy–force loss, as detailed in the Appendix. Table 1 reports accuracy and calibration across the seven MD17 molecules. Several consistent trends emerge. First, Dropout generally degrades energy and force accuracy and produces poorly calibrated uncertainty estimates. The diffusion-based CARD approach maintains the backbone's predictive accuracy because it reuses the point estimate for its mean prediction. However, its calibration remains weak across both MCal and RCal. In contrast, the MER and EMER (without Dropout) provide the best overall predictive accuracy. For energy, MER and EMER achieve the lowest MAE on most molecules. Crucially, for the main task of force prediction, the equivariant variants—EMER and EMER-Dropout—consistently perform best, achieving the lowest Force MAE on 5 out of 7 molecules, highlighting the advantage of preserving equivariance in the uncertainty parameterisation. In terms of calibration, MER-Dropout and EMER-Dropout yield the strongest calibration, demonstrating the benefit of the NIW prior. However, we stress that the apparently strong component-wise calibration (RCal) of non-equivariant MER-Dropout is misleading: because MD17 training and test samples come from the same distribution of orientations, therefore it is not robust and would collapses under distributional shifts (e.g. rotations).

| Mol | Metric | Backbone | Equivariant | | Nonequivariant | | Ours | | |
|---|---|---|---|---|---|---|---|---|---|
| | | | Ens. | Drop. | CARD | MER | EMER | MER-D | EMER-D |
| Aspirin | Energy MAE | 0.2197 | 0.1873 | 0.3549 | 0.2197 | **0.1926** | 0.1935 | 0.4600 | 0.3848 |
| | Force MAE | 0.5184 | 0.3799 | 0.4952 | 0.5184 | 0.4422 | 0.4217 | 0.4128 | **0.3924** |
| | MCal | – | 0.8167 | 0.9933 | 1.0810 | 1.4016 | 1.4291 | **0.2138** | 0.2414 |
| | Max RCal | – | 0.0801 | 0.4476 | 0.7790 | 0.9235 | 0.9620 | **0.1151** | 0.1306 |
| | Max NLL | – | 0.8120 | 2.2683 | 6.9817 | 6.2573 | 8.8739 | 0.7940 | **0.7334** |
| Ethanol | Energy MAE | 0.0872 | 0.0670 | 0.1801 | 0.0872 | 0.0682 | **0.0669** | 0.1140 | 0.1045 |
| | Force MAE | 0.3210 | 0.2364 | 0.3227 | 0.3210 | 0.2812 | 0.2843 | **0.2755** | 0.2843 |
| | MCal | – | 0.6743 | 0.8974 | 1.1185 | 1.4873 | 1.4879 | **0.2711** | 0.4060 |
| | Max RCal | – | 0.0692 | 0.3698 | 0.7430 | 1.0453 | 1.0259 | **0.1466** | 0.2314 |
| | Max NLL | – | 0.2507 | 1.7626 | 10.140 | 16.324 | 28.159 | **0.5150** | 0.5301 |
| Nap | Energy MAE | 0.1249 | 0.1168 | 0.3017 | 0.1249 | **0.1124** | 0.1153 | 0.3249 | 0.2309 |
| | Force MAE | 0.1296 | 0.0904 | 0.1520 | 0.1296 | 0.1036 | **0.1028** | 0.1517 | 0.1564 |
| | MCal | – | 0.7972 | 0.3846 | 0.4395 | 0.8860 | 0.9133 | 0.2047 | **0.1587** |
| | Max RCal | – | 0.0779 | 0.0768 | 0.3752 | 0.5049 | 0.4941 | **0.1005** | 0.1578 |
| | Max NLL | – | -0.6304 | 0.1107 | 1.0555 | -0.1228 | 0.4804 | **-0.4117** | -0.1836 |
| MDA | Energy MAE | 0.1183 | 0.1127 | 0.2214 | 0.1183 | 0.1038 | **0.1008** | 0.1479 | 0.1419 |
| | Force MAE | 0.4900 | 0.3790 | 0.4698 | 0.4900 | 0.4132 | 0.4286 | 0.4099 | **0.4008** |
| | MCal | – | 0.7626 | 1.0735 | 1.3320 | 1.4788 | 1.5250 | **0.4549** | 0.5317 |
| | Max RCal | – | 0.0714 | 0.5027 | 0.9373 | 1.0455 | 1.1097 | **0.2610** | 0.3213 |
| | Max NLL | – | 0.8031 | 2.7720 | 10.945 | 15.326 | 32.112 | 1.1694 | **1.1267** |
| SAC | Energy MAE | 0.1477 | 0.1262 | 0.2727 | 0.1477 | **0.1228** | 0.1375 | 0.2544 | 0.2495 |
| | Force MAE | 0.3286 | 0.2511 | 0.3455 | 0.3286 | **0.2564** | 0.2660 | 0.2729 | 0.2640 |
| | MCal | – | 0.7943 | 0.7863 | 0.9283 | 1.2957 | 1.2760 | 0.1711 | **0.1331** |
| | Max RCal | – | 0.0817 | 0.3207 | 0.7549 | 0.8115 | 0.8608 | 0.1722 | **0.1269** |
| | Max NLL | – | 0.3554 | 1.4537 | 6.4255 | 3.7537 | 5.2171 | 1.1807 | **0.4612** |
| Uracil | Energy MAE | 0.1123 | 0.1057 | 0.2517 | 0.1195 | **0.1052** | 0.1123 | 0.1998 | 0.1757 |
| | Force MAE | 0.2251 | 0.1660 | 0.2589 | 0.2251 | 0.1811 | **0.1778** | 0.2127 | 0.1991 |
| | MCal | – | 0.7527 | 0.4784 | 0.7958 | 1.1957 | 1.1884 | 0.3718 | 0.4209 |
| | Max RCal | – | 0.0661 | 0.1012 | 0.6422 | 0.7314 | 0.7981 | **0.1798** | 0.2399 |
| | Max NLL | – | -0.0920 | 0.7718 | 4.6010 | 2.5799 | 3.2327 | **-0.0436** | 0.1669 |
| Toluene | Energy MAE | 0.1018 | 0.0929 | 0.2586 | 0.1018 | 0.0983 | **0.0951** | 0.2397 | 0.2798 |
| | Force MAE | 0.1511 | 0.1084 | 0.1681 | 0.1511 | 0.1237 | **0.1200** | 0.1568 | 0.1567 |
| | MCal | – | 0.8090 | 0.4373 | 0.5489 | 1.0660 | 1.0756 | 0.2084 | **0.1764** |
| | Max RCal | – | 0.0749 | 0.0726 | 0.4406 | 0.0634 | 0.6351 | 0.1089 | **0.0716** |
| | Max NLL | – | -0.4513 | 0.2003 | 1.9618 | 1.1105 | 1.1232 | **-0.2478** | -0.2299 |

Table 1: Performance metrics on the MD17 dataset. Calibration and NLL are evaluated on atomic forces only. Bold indicates the best single-model performance. MER-D and EMER-D denote MER-Dropout and EMER-Dropout, respectively. Our proposed methods (EMER, MER-D, and EMER-D) achieve the best accuracy and calibration across the majority of tasks.

To validate this hypothesis, we further evaluated our method on the Naphthalene and Uracil subsets of MD17, which exhibit highly anisotropic force distributions. For these two molecules, the force vectors lie predominantly within the $xy$-plane (see Figure 5 in the Appendix). We applied a coordinate rotation to the test set to align these vectors closer to the $z$-axis. Table 2 reports the Force MAE, MCal, and dimensional RCal before and after rotation. We observe that while EMER-Dropout and MER-Dropout yield comparable metrics on the unrotated case, MER-Dropout fails after rotation. Specifically, the RCal for MER-Dropout spikes from $\sim 0.15$ to $0.70$ on the rotated set. In contrast, EMER-Dropout exhibits a significantly more moderate increase (from $\sim 0.15$ to $0.40$), thereby maintaining much greater stability. This contrast underscores the critical importance of preserving equivariance in vector UE.

For OC20, we employ a direct force output head rather than computing the negative energy gradient, as the later one is computationally prohibitive for larger graphs. The performance on the OC20

Table 2: Calibration stability under rotation for Naphthalene and Uracil (MD17). Due to anisotropic force distributions, the non-equivariant MER-Dropout exhibits significant degradation in RCal upon rotation, particularly along the $z$-axis. EMER-Dropout mitigates this rotational sensitivity compared to the non-equivariant MER-Dropout, while the Ensemble method maintains gold standard component-wise calibration.

| Mol. | Method | Energy MAE | Force MAE | MCal | RCal w/o rotation | | | RCal w/ rotation | | |
|---|---|---|---|---|---|---|---|---|---|---|
| | | | | | $x$ | $y$ | $z$ | $x$ | $y$ | $z$ |
| Nap | Ensemble | 0.117 | 0.090 | 0.797 | 0.060 | 0.058 | 0.163 | 0.084 | 0.102 | 0.061 |
| | EMER-Dropout | 0.231 | 0.156 | 0.221 | 0.120 | 0.150 | 0.158 | 0.432 | 0.462 | 0.422 |
| | MER-Dropout | 0.325 | 0.152 | 0.205 | 0.128 | 0.107 | 0.124 | 0.665 | 0.677 | 0.883 |
| Uracil | Ensemble | 0.106 | 0.166 | 0.753 | 0.064 | 0.076 | 0.136 | 0.059 | 0.063 | 0.073 |
| | EMER-Dropout | 0.176 | 0.199 | 0.421 | 0.240 | 0.219 | 0.189 | 0.376 | 0.371 | 0.099 |
| | MER-Dropout | 0.200 | 0.213 | 0.372 | 0.214 | 0.182 | 0.142 | 0.625 | 0.650 | 0.781 |

| Split | Metric | Backbone | Equivariant | | Nonequivariant | | Ours | | |
|---|---|---|---|---|---|---|---|---|---|
| | | | Ens. | Drop. | CARD | MER | EMER | MER-D | EMER-D |
| Val ID | Energy MAE | 0.4745 | 0.4210 | **0.4567** | 0.4745 | 0.6216 | 0.5806 | 0.6293 | 0.5047 |
| | Force MAE | 0.0710 | 0.0590 | 0.0723 | 0.0710 | 0.0815 | 0.0636 | 0.0807 | **0.0628** |
| | MCal | – | 1.3703 | 1.4320 | 0.4614 | 0.4464 | 0.6920 | **0.2727** | 0.2955 |
| | Max RCal | | 0.7788 | 0.9663 | 0.4673 | 0.2687 | 0.3992 | **0.1612** | 0.1776 |
| | Max NLL | | 14.1246 | 31.450 | **-0.6602** | 11.9579 | 1.0776 | 0.6695 | -0.0733 |
| Val OOD Ads | Energy MAE | 0.6192 | 0.5662 | 0.6094 | 0.6192 | 0.6315 | 0.5809 | 0.6209 | **0.5211** |
| | Force MAE | 0.0730 | 0.0630 | 0.0737 | 0.0730 | 0.0810 | 0.0740 | 0.0839 | **0.0707** |
| | MCal | – | 1.3724 | 1.4364 | 0.3785 | 0.6035 | 0.9048 | **0.3566** | 0.4927 |
| | Max RCal | | 0.7774 | 0.9672 | 0.4394 | 0.2538 | 0.5609 | **0.2067** | 0.3077 |
| | Max NLL | | 14.603 | 30.882 | **-0.4712** | 12.1289 | 2.7852 | 0.8275 | 0.3941 |
| Val OOD Cat | Energy MAE | 0.5293 | 0.4690 | **0.4992** | 0.5293 | 0.6707 | 0.6525 | 0.6973 | 0.5913 |
| | Force MAE | 0.0816 | 0.0676 | 0.0799 | 0.0816 | 0.0747 | **0.0642** | 0.0810 | 0.0647 |
| | MCal | – | 1.3244 | 1.4678 | 0.4873 | 0.4493 | 0.7096 | **0.2868** | 0.3108 |
| | Max RCal | | 0.7848 | 0.9699 | 0.5100 | 0.2859 | 0.4149 | **0.1666** | 0.1912 |
| | Max NLL | | 15.7110 | 31.563 | **-0.5757** | 7.3281 | 1.3912 | 0.7056 | -0.1668 |
| Val OOD Both | Energy MAE | 0.7515 | 0.6902 | 0.7300 | 0.7515 | 0.8254 | 0.7856 | 0.8441 | **0.7055** |
| | Force MAE | 0.0906 | 0.0624 | 0.0887 | 0.0906 | 0.0926 | 0.0850 | 0.0953 | **0.0811** |
| | MCal | – | 1.3499 | 1.4939 | **0.3280** | 0.6667 | 0.9737 | 0.4094 | 0.5515 |
| | Max RCal | | 0.7921 | 0.9701 | 0.3676 | 0.4189 | 0.6107 | **0.2519** | 0.3494 |
| | Max NLL | | 19.4712 | 34.664 | **-0.0605** | 29.4542 | 4.0050 | 1.4568 | 0.9357 |

Table 3: Performance metrics for different splits with the OC20 dataset. Calibration and NLL are evaluated on atomic forces only. Bold indicates the best single-model performance. MER-D and EMER-D denote MER-Dropout and EMER-Dropout, respectively. While Dropout baseline attains the lowest energy MAE occasionaly, our proposed equivariant method, EMER and EMER-Dropout obtained the lowest force MAE for all splits. MER-D and EMER-D also demonstrate best calibration in both MCal and Max RCal for the majority of tasks.

is summarized in Table 3 across four validation splits: In-Domain (ID), Out-of-Domain Adsorbate (OOD Ads), OOD Catalyst (OOD Cat), and OOD Both. For energy prediction, the Dropout baseline remains competitive, achieving the lowest MAE on two splits, while EMER-Dropout secures the best on the remaining two. However, the benefits of our equivariant approach are pronounced in the more relevant force prediction: EMER-Dropout attains the highest accuracy on the ID, OOD Ads, and OOD Both splits, while EMER leads on OOD Cat. Regarding calibration, MER-Dropout generally yields the best MCal and Max RCal scores, while CARD achieves the lowest NLL. Yet, as demonstrated in Table 2, the apparent calibration success of non-equivariant methods stems from the similarity in marginal force distributions between the training and test sets. While EMER-Dropout exhibits marginally higher calibration error than MER-Dropout (e.g., MCal of 0.295 vs. 0.273 for Val ID), it ensures rotational robustness, which is essential for physical consistency.

We further evaluate our method on QM7X for dipole-moment prediction using three backbones (PaiNN, NequIP (Batzner et al., 2022), and MACE (Batatia et al., 2022)), with results reported in Table 4. The performance is very similar to force prediction task – MER attains the lowest MAE for two backbones, and MER-Dropout achieves the best MAE on the remaining one; overall, their MAEs are even better than Ensemble. However, raw MER and EMER's UE are poorly calibrated. Introducing dropout to regulaziring the latent vectors (MER-Dropout and EMER-Dropout) markedly

| Model | Metric | Backbone | Equivariant | | Nonequivariant | | Ours | | |
|---|---|---|---|---|---|---|---|---|---|
| | | | Ens. | Drop. | CARD | MER | EMER | MER-D | EMER-D |
| PaiNN | DP MAE | 0.0073 | 0.0054 | 0.0068 | 0.0073 | **0.0053** | 0.0056 | 0.0060 | 0.0058 |
| | MCal | – | 0.4394 | 1.0030 | 1.6881 | 1.5647 | 1.6011 | 0.4922 | **0.4523** |
| | Max RCal | – | 0.1160 | 0.1723 | 1.2034 | 1.1861 | 1.2607 | 0.3214 | **0.3058** |
| | Max NLL | – | -3.5406 | -2.7610 | -2.6909 | 44.234 | 54.967 | -3.0513 | **-3.1206** |
| NequIP | DP MAE | 0.0069 | 0.0060 | 0.0079 | 0.0069 | **0.0056** | 0.0059 | 0.0062 | 0.0060 |
| | MCal | – | 0.5122 | 0.6117 | 1.4434 | 1.2597 | 1.3868 | 0.4437 | **0.2019** |
| | Max RCal | – | 0.1866 | 0.1155 | 1.1938 | 0.7871 | 0.7990 | 0.3070 | **0.1388** |
| | Max NLL | – | -3.1946 | -2.9163 | -2.7945 | -0.1381 | 3.7854 | **-3.5205** | -3.4627 |
| MACE | DP MAE | 0.0079 | 0.0074 | 0.0084 | 0.0079 | 0.0044 | 0.0043 | **0.0039** | 0.0041 |
| | MCal | – | 0.6728 | 1.1019 | 0.7507 | 1.5332 | 1.4549 | **0.7417** | 0.8728 |
| | Max RCal | – | 0.2816 | 0.5373 | 0.6256 | 1.1269 | 1.0334 | **0.4634** | 0.5267 |
| | Max NLL | – | -2.9139 | -1.3933 | -1.1121 | 24.986 | 20.272 | -2.7153 | **-2.7868** |

Table 4: Performance metrics for dipole moment prediction using QM7X dataset. Bold indicates the best single-model performance. MER-D and EMER-D denote MER-Dropout and EMER-Dropout, respectively. Our proposed methods (EMER, MER-D, and EMER-D) achieve the best accuracy and calibration across the majority of tasks.

Table 5: Ablation study on the design of the matrix $\boldsymbol{\Psi} = \exp(\alpha) + \boldsymbol{F}\boldsymbol{F}^\top$. All models use PaiNN as a backbone. We compare configurations with and without the scalar term $\exp(\alpha)$ and vary the dimensionality $m \in \{0, 1, 2, 3\}$ of the matrix $\boldsymbol{F} \in \mathbb{R}^{3 \times m}$, testing each setup with and without dropout. Dropout ratio are set to be 0.01 and 0.1, for MD17 and QM7X, respectively. The results demonstrate that the $\exp(\alpha)$ term is essential for stability. Increasing the dimensionality to $m = 3$ also leads to instability when dropout is applied.

| | MD17+EMER | | | MD17+EMER Dropout | | | QM7X+EMER | | QM7X+EMER Dropout | |
| Model | EMAE | FMAE | MCal | EMAE | FMAE | MCal | DPMAE | MCal | DPMAE | MCal |
|---|---|---|---|---|---|---|---|---|---|---|
| w/o $\alpha, m = 3$ | - | - | - | | - | - | 0.0053 | 1.107 | - | - |
| w/ $\alpha, m = 0$ | 0.119 | 0.262 | 1.069 | 0.248 | 0.276 | 0.246 | 0.0054 | 1.553 | 0.0057 | 0.4737 |
| w/ $\alpha, m = 1$ | 0.118 | 0.259 | 1.267 | 0.206 | 0.270 | 0.298 | 0.0055 | 1.560 | 0.0063 | 0.4544 |
| w/ $\alpha, m = 2$ | 0.119 | 0.250 | 1.283 | 0.221 | 0.264 | 0.308 | 0.0055 | 1.601 | 0.0058 | 0.4523 |
| w/ $\alpha, m = 3$ | 0.120 | 0.252 | 1.291 | 0.209 | 0.266 | 0.293 | 0.0055 | 1.584 | - | - |

improves calibration, yielding the lowest score for all backbones, with only a slight reduction in accuracy; even so, its MAE remains better than baseline models trained with MSE or L1 losses.

## 4.1 ABLATION STUDY

An ablation study was conducted to evaluate the design of the matrix $\boldsymbol{\Psi} = \exp(\alpha) + \boldsymbol{F}\boldsymbol{F}^\top$. Using PaiNN as the backbone, we compared configurations with and without the $\exp(\alpha)$ term and varied the dimensionality $m$ of the matrix $\boldsymbol{F} \in \mathbb{R}^{3 \times m}$. These configurations were tested with and without dropout on the MD17 and QM7X benchmarks, with results summarized in Table 5. The model variant excluding the $\exp(\alpha)$ term was unstable during training on both MD17 and QM7X with dropout. In contrast, models that included the term trained stably and achieved comparable performance across all metrics for $m \in 0, 1, 2, 3$. The configuration with $m = 0$ makes the covariance matrix to be an isotropic diagonal matrix. While this yields good MCal (likely because the Mahalanobis distance can be minimized by a uniform uncertainty sphere), it inherently fails to capture directional uncertainties, limiting its physical utility. Increasing the dimensionality from $m = 1$ to $m = 3$ led to instability on QM7X+EMER Dropout, supporting our hypothesis that $m = 3$ is overparameterized. To further investigate the trade-off between parameter efficiency and model expressiveness, we analyzed the performance difference between $m = 1$ and $m = 2$. While $m = 1$ and $m = 2$ yielded comparable performance on standard metrics, we evaluated their robustness on molecules with anisotropic distribution, specifically Nap and Uracil (Table 11). For them a rank-1 parameterization ($m = 1$) lacked the expressiveness to decouple in-plane and out-of-plane uncertainties, resulting in poor RCal along the $z$-axis. In contrast, $m = 2$ maintained robust calibration across all dimensions. Consequently, we adopted $m = 2$. This decision is supported by a singular value decomposition of the learned matrix (Figure 4), which shows the sum of the first two singular values accounts for $> 90\%$ of the variance, confirming that a rank-2 approximation is sufficient to capture physical symmetries.

## 5 CONCLUSION

In this work, we addressed the critical and previously overlooked challenge of obtaining reliable, geometrically consistent uncertainty estimates for equivariant vector-valued properties in the physical sciences. We demonstrated that conventional UE methods fail to respect underlying physical symmetries, yielding uncertainty measures that are not only poorly calibrated but also physically meaningless. To resolve this, we introduced Equivariant Multivariate Evidential Regression (EMER), a novel framework built upon a new parameterization for the covariance matrix that is provably equivariant, positive definite, and identifiable. Furthermore, we proposed a Mahalanobis distance-based metric to enable the rigorous evaluation of multivariate uncertainty calibration.

Our extensive experiments on standard molecular property prediction benchmarks—MD17, OC20, and QM7-X—empirically validate our theoretical contributions. The results consistently show that EMER outperforms established baselines, achieving a new state of the art in both predictive accuracy and uncertainty calibration for force and dipole moment prediction. This work represents a significant step toward building more reliable and trustworthy machine learning models for scientific applications. By providing a principled and practical approach to equivariant UE, our framework paves the way for more robust active learning workflows, error detection in large-scale simulations, and adaptive multiscale modeling, thereby accelerating the pace of scientific discovery.

ETHICS STATEMENT

This research focuses on the development of a foundational machine learning methodology for uncertainty estimation in the physical sciences. Our work is theoretical and algorithmic in nature, and we do not foresee any direct negative societal impacts or ethical concerns. The datasets used in our experiments—MD17, OC20, and QM7-X—are established public benchmarks for molecular property prediction and do not contain personally identifiable or sensitive information. The proposed methods are intended to improve the reliability of machine learning models for scientific discovery and do not involve human subjects, raise privacy issues, or have immediate dual-use applications.

REPRODUCIBILITY STATEMENT

To ensure the reproducibility of our results, we will release the complete source code, including scripts for data processing, model training, and evaluation, upon publication. Our paper provides a detailed description of the proposed equivariant multivariate evidential regression framework. All experiments were conducted on publicly available datasets (MD17, OC20, QM7-X). The appendix contains a comprehensive description of the experimental setup, including dataset splits, model hyperparameters, and implementation details for all baselines, which will allow for a faithful reproduction of our findings.

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

# A    APPENDIX

## A.1    PROOF OF PROPOSITION 3.1

We subsequently prove the three satisfied constraints.

**Positive Definiteness (PD):** Since $\alpha \in \mathbb{R}$, $\exp(\alpha) > 0$. The first term, $\exp(\alpha)\mathbf{I}$, is a PD matrix. The second term, $\boldsymbol{F}\boldsymbol{F}^\top$, is a PSD matrix. As the sum of a PD matrix and a PSD matrix is always PD, $\boldsymbol{\Psi}$ is guaranteed to be PD.

**Equivariance:** The equivariant nature of $\boldsymbol{F}$ means an input rotation $\mathbf{R}$ generates a new matrix $\boldsymbol{F}' = \mathbf{R}\boldsymbol{F}$, while the invariant scalar $\alpha$ remains unchanged. The resulting scale matrix $\boldsymbol{\Psi}'$ then correctly transforms under rotation:

$$\boldsymbol{\Psi}' = \exp(\alpha)\mathbf{I} + \boldsymbol{F}'(\boldsymbol{F}')^\top = \exp(\alpha)\mathbf{I} + (\mathbf{R}\boldsymbol{F})(\mathbf{R}\boldsymbol{F})^\top = \mathbf{R}\left(\exp(\alpha)\mathbf{I} + \boldsymbol{F}\boldsymbol{F}^\top\right)\mathbf{R}^\top = \mathbf{R}\boldsymbol{\Psi}\mathbf{R}^\top$$

This demonstrates that the proposed parameterization satisfies the equivariance property.

**Sign-Flip Identifiability:** Assuming $\boldsymbol{\Psi}_1 = \boldsymbol{\Psi}_2$, we analyze the eigenspectra. $\boldsymbol{\Psi}$ has degenerate minimum eigenvalues $\lambda_{\min} = \exp(\alpha)$ and a unique largest eigenvalue $\lambda_{\max} = \exp(\alpha) + \|\boldsymbol{F}\|^2$. Equating the degenerate eigenvalues forces $\exp(\alpha_1) = \exp(\alpha_2)$, which implies $\alpha_1 = \alpha_2$. Equating the largest eigenvalues then yields $\|\boldsymbol{F}_1\|^2 = \|\boldsymbol{F}_2\|^2$. Since the principal eigenvector corresponding to $\lambda_{\max}$ is proportional to $\boldsymbol{F}$, the constraint $\boldsymbol{\Psi}_1 = \boldsymbol{\Psi}_2$ ensures $\boldsymbol{F}_1$ and $\boldsymbol{F}_2$ are collinear with equal norms, thus $\boldsymbol{F_1} = \pm\boldsymbol{F_2}$. The parameterization is identifiable up to the sign of $\boldsymbol{F}$.

## A.2    IMPLEMENTATION DETAILS

**Backbones** Equivariant graph neural networks were employed due to their vector-prediction capabilities. Specifically, this study utilizes PaiNN (Schütt et al., 2021), NequIP (Batzner et al., 2022), and MACE (Batatia et al., 2022). All models were trained using the default hyperparameters from their respective original works, with two exceptions. First, for PaiNN on OC20, the number of interaction blocks was set to 4 and the hidden dimension to 512. Second, to improve baseline accuracy on QM7X, the hidden irreps for NequIP and MACE were increased to `64x0o+64x0e+64x1o+64x1e+64x2o+64x2e` and `128x0e+128x1o+128x2e`, respectively.

The backbones are trained using a standard combined energy and force loss function:

$$\mathcal{L} = (1 - \rho)\,\mathcal{L}_{\text{energy}} + \rho\mathcal{L}_{\text{force}} \tag{6}$$

where $\mathcal{L}_{\text{energy}}$ is the energy loss for the total energy (or adsorption enenergy for OC20), $\mathcal{L}_{\text{force}}$ is the force loss, and $\rho \in [0, 1]$ is a weighting coefficient that balances the contribution of the two terms. Specifically, the weighting coefficient $\rho$ and the choice of loss functions for the energy ($\mathcal{L}_{\text{energy}}$) and force ($\mathcal{L}_{\text{force}}$) terms are detailed in Table 6 for the MD17 and OC20 datasets. For the task of dipole moment prediction on the QM7X dataset, the MER loss was employed for the MER and EMER families, L2 loss was used for PaiNN and L1 loss was utilized for NequIP and MACE.

Table 6: Loss function hyperparameters for different benchmarks

| Dataset & Method | Weighting $\rho$ | $\mathcal{L}_{\text{energy}}$ | $\mathcal{L}_{\text{force}}$ |
|---|---|---|---|
| MD17 | 0.99 | L1 | L2 |
| MD17+EMER | 0.99 | L1 | MER |
| OC20 | 0.96 | L1 | L1 |
| OC20+EMER | 0.96 | L1 | MER |

Table 7: Hyperparameters of datasets used with all models

| | MD17 | OC20 | QM7X |
|---|---|---|---|
| batch size | 10 | 32 | 32 |
| num of train | 1000 | 90000 | 5584 |
| num of test | 1000 | 10000 | 698 |

**Vector Output Head Details** For the MD17 dataset, forces are computed as the negative gradient of the predicted energy. In contrast, for the OC20 dataset, forces are predicted directly using an equivariant vector output head.

To predict dipole moments, we follow the approach of Schütt et al. (2021), given by

$$\vec{\mu} = \sum_{i=1}^{N} \left( \vec{\mu}_{\text{atom}}(\vec{v}_i) + q_{\text{atom}}(s_i)\vec{r}_i \right),$$

where $\vec{\mu}_{\text{atom}}$ is the local atomic dipole, $q_{\text{atom}}$ is the learned partial atomic charge, and $\vec{r}_i$ is the atomic position vector. This formulation constructs the total dipole moment by summing contributions from both induced atomic dipoles and the dipoles generated by the learned atomic charges.

**Monte Carlo Dropout** Bernoulli Dropout is employed for MC Dropout uncertainty estimation. It is exclusively applied to the features following each message-passing interaction block. Crucially, no dropout is utilized within the interaction blocks or the final readout layers. To maintain $SO(3)$-equivariance, we apply Equivariant Dropout (Liao & Smidt, 2023; Liao et al., 2024) to all tensorial features ($l \geq 0$), ensuring that the transformation properties of the representations are preserved. The hyperrameters used for MD Dropout and the backbone are listed in Table 8 and Table 9.

Table 8: Hyperparameters of PaiNN

| Model / Hyperparameter | MD17 | OC20 |
|---|---|---|
| *PaiNN* | | |
| learning rate | 0.001 | 0.00025 |
| warmup steps | 500 | 20000 |
| decay steps | 40000 | 2000000 |
| decay rate | 0.01 | 0.01 |
| EMA decay | 0.99 | 0.99 |
| Epochs | 400 | 120 |
| Batch size | 10 | 32 |
| force weighting factor $\rho$ | 0.99 | 0.96 |
| *PaiNN Dropout* | | |
| learning rate | 0.001 | 0.00025 |
| warmup steps | 500 | 20000 |
| decay steps | 40000 | 2000000 |
| decay rate | 0.01 | 0.01 |
| EMA decay | 0.99 | 0.99 |
| Epochs | 400 | 120 |
| Batch size | 10 | 32 |
| force weighting factor $\rho$ | 0.99 | 0.96 |
| dropout ratio | 0.01 | 0.1 |
| dropout locations | after every interaction block | |

**Classification and Regression Diffusion** We implemented the Classification and Regression Diffusion (CARD) model (Han et al., 2022) as a conditional generative head attached to pre-trained equivariant backbones (e.g., PaiNN, MACE). Since CARD is originally designed for scalar regression, we adapted it to vector-valued prediction by flattening the $(N, 3, m)$ tensor of $l = 1$ irreps into a 2D tensor of shape $(N \times 3, m)$, where each of the $x$, $y$, and $z$ force components is treated as an independent scalar variable. This adaptation effectively removes the coupling between vector components, meaning CARD receives no information about rotational structure beyond the backbone's atomic embeddings and its deterministic force prediction.

The diffusion process uses 1,000 timesteps with a linear beta schedule from $\beta_{\text{start}} = 10^{-4}$ to $\beta_{\text{end}} = 0.02$. The model is trained for 100 epochs using Adam (learning rate $10^{-3}$, batch size 256), minimizing the mean-squared error between the predicted and true noise. Gradient norms are clipped to 1.0.

Table 9: Hyperparameters of PaiNN, NequIP, MACE for QM7X dipole moment prediction

| Hyperparameter | PaiNN | NequIP | MACE |
|---|---|---|---|
| learning rate | 0.001 | 0.01 | 0.01 |
| warmup epochs | 1 | 1 | 1 |
| decay epochs | 200 | 250 | 300 |
| decay rate | 0.01 | 0.01 | 0.01 |
| EMA decay | 0.99 | 0.99 | 0.99 |
| Epochs | 200 | 250 | 300 |
| Batch size | 10 | 32 | 32 |
| *Dropout* | | | |
| learning rate | 0.001 | 0.01 | 0.01 |
| warmup epochs | 1 | 1 | 1 |
| decay epochs | 200 | 250 | 300 |
| decay rate | 0.01 | 0.01 | 0.01 |
| EMA decay | 0.99 | 0.99 | 0.99 |
| Epochs | 200 | 250 | 300 |
| Batch size | 10 | 32 | 32 |
| dropout ratio $\rho$ | 0.1 | 0.1 | 0.1 |
| dropout locations | after every interaction block | | |

For uncertainty estimation, inference is performed via Monte Carlo reverse diffusion with $K = 100$ samples per atom, starting from standard Gaussian noise. The predictive mean is computed as the sample average, and uncertainty is quantified by the sample variance. Because the model predicts each coordinate independently, the estimated covariance matrix is strictly diagonal. Consequently, this scalarized CARD adaptation cannot model cross-coordinate correlations and breaks rotational equivariance.

### A.3 EVALUATION METRICS

To rigorously assess the reliability of our proposed framework, we employ three complementary metrics: the Mahalanobis Calibration Score (MCal) for vector-valued consistency proposed in this work, the component-wise Raw Calibration Score (RCal) for marginal consistency, and the Negative Log-Likelihood (NLL) for overall predictive distribution quality.

**Mahalanobis Calibration Score (MCal)** Standard calibration metrics designed for scalar regression are ill-suited for equivariant vector quantities, as they fail to account for correlations between components and rotational symmetries. Mahalanobis Calibration Score is designed to address this problem. For a predicted mean vector $\boldsymbol{\mu}_i$ and covariance matrix $\boldsymbol{\Sigma}_i$ corresponding to the ground truth $\mathbf{v}_{\text{true}}$, the squared Mahalanobis distance is defined as:

$$d_i^2 = (\mathbf{v}_{\text{true}} - \boldsymbol{\mu}_i)^\top \boldsymbol{\Sigma}_i^{-1} (\mathbf{v}_{\text{true}} - \boldsymbol{\mu}_i)$$

If the model is perfectly calibrated and the predictive distribution is multivariate normal, the squared Mahalanobis distances $\{d_i^2\}_{i=1}^N$ should follow a Chi-squared distribution with $n$ degrees of freedom ($\chi_n^2$), where $n$ is the dimensionality of the vector (here, $n = 3$). We define the calibration score by comparing the empirical cumulative distribution of the observed distances against the theoretical CDF of the $\chi_n^2$ distribution. Let $F_{\chi_n^2}(z)$ be the theoretical CDF evaluated at threshold $z$. We compute the empirical frequency of observed distances falling below the theoretical quantile associated with probability level $p$:

$$\hat{p}(p) = \frac{1}{N} \sum_{i=1}^N \mathbb{I}\left[ d_i^2 \leq F_{\chi_n^2}^{-1}(p) \right]$$

The MCal score is then calculated as the root mean squared error (RMSE) between the empirical frequency and the expected probability levels $p \in \{0.1, \ldots, 0.9\}$:

$$\text{MCal} = \sqrt{\frac{1}{M} \sum_{j=0}^{M} (\hat{p}(p_j) - p_j)^2}$$

A lower MCal indicates that the model's multivariate uncertainty ellipsoids correctly encapsulate the ground truth with the expected frequency.

**Component-wise Raw Calibration Score (RCal)** To facilitate comparison with methods that treat vector components independently, we also report the Raw Calibration Score (RCal) in Wollschläger et al. (2023), originally defined by Kuleshov et al. (2018a). The empirical probability is computed as the fraction of test samples where the true scalar value falls within this interval:

$$\hat{p}_{\text{marg}}(p) = \frac{1}{N} \sum_{i=1}^{N} \mathbb{I}\left[ y_{i,d} \in I_p^{(i,d)} \right]$$

where $y_{i,d}$ is the $d$-th component of the $i$-th target vector. The scalar calibration score is defined as:

$$\text{RCal}_d = \sqrt{\sum_{p \in \{0.1, \ldots, 0.9\}} (\hat{p}_{\text{marg}}(p) - p)^2}$$

Since an equivariant vector model may exhibit different calibration behaviors along different axes (though ideally, it should not), reporting the average RCal can be misleading. Therefore, we report the Max RCal, which is the maximum calibration error observed across the three spatial dimensions: Max RCal $= \max_{d \in \{x,y,z\}}(\text{RCal}_d)$. This serves as a worst-case estimate of marginal calibration.

**Negative Log-Likelihood (NLL)** While calibration scores assess the alignment of uncertainty estimates, they do not account for the sharpness (precision) of the predictions. The Negative Log-Likelihood (NLL) is a proper scoring rule that balances predictive accuracy with uncertainty quantification. For a univariate Gaussian predictive distribution $\mathcal{N}(\mu, \sigma^2)$, the NLL for a single scalar observation is given by:

$$\text{NLL}(y, \mu, \sigma^2) = \frac{1}{2} \log(\sigma^2) + \frac{(y - \mu)^2}{2\sigma^2} + \frac{1}{2} \log(2\pi)$$

Similar to RCal, we calculate the NLL independently for each dimension and report the maximum NLL over dimensions to capture the worst-case performance.

### A.4 ADDITIONAL EXPERIMENTS

**Dropout Ratio Ablation Study** We investigated the effect of the dropout ratio, a key hyperparameter in our model, by evaluating values of 0.01, 0.05, 0.1, and 0.2 using PaiNN backbone. The results, summarized in Table 10, show differing trends across the evaluation datasets. For the MD17 benchmark, a clear inverse relationship is observed, where increasing the dropout ratio leads to a degradation in performance across energy MAE, force MAE, and the force calibration score. Conversely, on the QM7X dataset, the trend is less uniform; while the lowest dropout ratio (0.01) yields the best dipole moment MAE, the dipole calibration improves as the dropout ratio increases. Prioritizing the performance on energy and force prediction, which are central to MD simulations, we selected a dropout ratio of 0.01 for our final model, as it provides the best overall accuracy on the MD17 tasks. For QM7X, we selected a dropout ratio of 0.1 for accuracy/calibration balance.

Table 10: Ablation study on the dropout ratio using PaiNN backbone

| Dropout Ratio | 0.01 | 0.05 | 0.1 | 0.2 |
|---|---|---|---|---|
| *MD17 Average* | | | | |
| Energy MAE | 0.2069 | 0.4404 | 0.6029 | 0.5720 |
| Force MAE | 0.2709 | 0.4244 | 0.5767 | 0.8758 |
| Force Calibration Score | 0.2989 | 0.5025 | 0.8201 | 1.1753 |
| *QM7X Dipole Moment* | | | | |
| DP MAE | 0.0056 | 0.0069 | 0.0063 | 0.0075 |
| DP Calibration | 0.5844 | 0.8068 | 0.4544 | 0.1757 |

**Marginal Calibration and Dimensionality Analysis.** Although $m = 1$ and $m = 2$ achieve similar predictive accuracy and global MCal in Table 5, we further evaluated their marginal calibration using Max RCal and Max NLL. As shown in Table 11, for the planar molecules Nap and Uracil from MD17, $m = 1$ leads to severe miscalibration along the $z$-axis (RCal of 0.487 and 0.490, respectively). In contrast, increasing $m$ to 2 effectively mitigates this issue, reducing the RCal to 0.158 and 0.189. These results indicate that $m = 2$ is necessary to maintain marginal calibration.

Table 11: Ablation study results showing performance metrics of EMER-Dropout with different $F$ dimension.

| Mol. | Method | Energy MAE | Force MAE | MCal | RCal w/o rotation | | |
|---|---|---|---|---|---|---|---|
| | | | | | $x$ | $y$ | $z$ |
| | Ensemble | 0.117 | 0.090 | 0.797 | 0.060 | 0.058 | 0.163 |
| Nap | EMER-D (m=1) | 0.167 | 0.146 | 0.221 | 0.100 | 0.082 | 0.487 |
| | EMER-D (m=2) | 0.231 | 0.156 | 0.221 | 0.120 | 0.150 | 0.158 |
| | Ensemble | 0.106 | 0.166 | 0.753 | 0.064 | 0.076 | 0.136 |
| Uracil | EMER-D (m=1) | 0.174 | 0.208 | 0.391 | 0.091 | 0.099 | 0.490 |
| | EMER-D (m=2) | 0.175 | 0.199 | 0.421 | 0.240 | 0.219 | 0.189 |

**Force Loss Weight** For machine learning force fields that jointly predict energy and forces, the weight assigned to the force loss is a critical hyperparameter. We evaluated the energy and force MAE across varying weights, as detailed in Table 12. The results indicate that a coefficient in the range of 0.99–0.999 minimizes force MAE, while a range of 0.8–0.99 yields the lowest energy MAE. This suggests that our choice of 0.99 provides a reasonable balance between energy and force accuracy.

Table 12: Ablation study of force weight for aspirin and ethanol on the MD17 dataset.

| Mol | | Force Loss Weight | | | | | |
|---|---|---|---|---|---|---|---|
| | | 0.01 | 0.2 | 0.5 | 0.8 | 0.99 | 0.999 |
| Aspirin | Energy MAE | 3.65 | 1.60 | 0.40 | 0.39 | 0.36 | 0.46 |
| | Force MAE | 19.48 | 4.74 | 0.69 | 0.51 | 0.42 | 0.40 |
| Ethanol | Energy MAE | 0.98 | 0.14 | 0.11 | 0.09 | 0.12 | 0.11 |
| | Force MAE | 7.60 | 0.65 | 0.43 | 0.33 | 0.29 | 0.28 |

**Active Learning.** We evaluated the performance of our proposed method in an active learning setting for QM7X dipole moment prediction. Motivated by the high cost of molecular verification in wet-lab environments, we employed a limited budget of 10 iterations, querying 20 samples per iteration based on the trace of the uncertainty covariance matrix. We benchmarked against single model based methods, including Dropout, MER. As shown in Figure 2, the proposed equivariant methods (EMER and EMER-Dropout) consistently achieved lower MAE compared to the baselines, underscoring the importance of incorporating equivariant UE for vector properties.

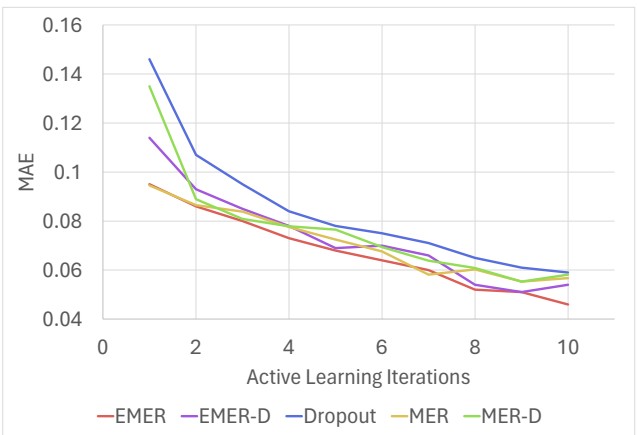

Figure 2: MAE of dipole moment prediction versus the number of active learning iterations. The proposed equivariant methods, EMER and EMER-Dropout, consistently yield the lowest error across all iterations.

### A.5 ADDITIONAL VISUALIZATION

Figure 3 shows the overiview of our proposed framed. Expcept for the general vector output head $\mu_0$, two additional heads are needed to output the parameters of Multivariate Evidential Loss. One is a scaler head which output scalers $\nu$ and $\alpha$, and another one is a vector head output a $3 \times n$ matrix $\boldsymbol{F}$.

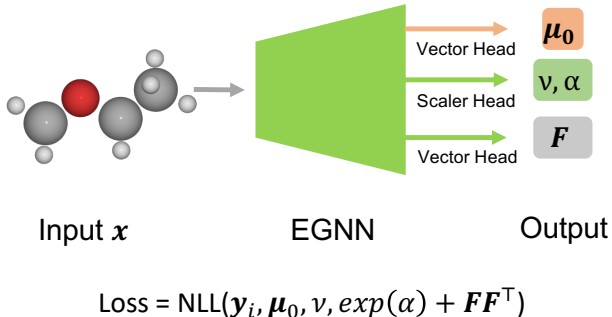

$$\text{Loss} = \text{NLL}(\boldsymbol{y}_i, \boldsymbol{\mu}_0, \nu, exp(\alpha) + \boldsymbol{F}\boldsymbol{F}^\top)$$

Figure 3: Overview of the proposed framework. Two additional uncertainty heads are incorporated to produce the evidential distribution parameters.

We performed a singular value decomposition of the learned matrix, as shown in Figure 4. It shows that the sum of the first two singular values accounts for $> 90\%$ of the variance, confirming that a rank-2 approximation is sufficient to capture physical symmetries without the computational cost of non-parametric methods.

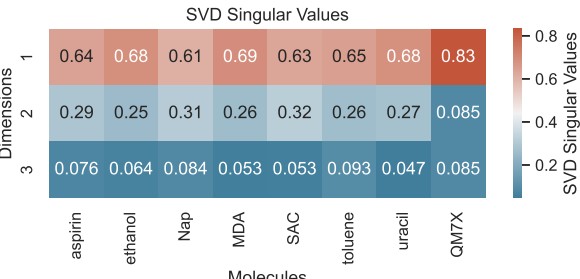

Figure 4: SVD singular values for molecules in MD17 and QM7X. For all molecules, the first two singular value is dominant, indicating that the learned matrix is well-approximated by a rank-2 component.

Figure 5 illustrates the distribution of force labels for Aspirin and Naphthalene (Nap). For Aspirin, the forces are uniformly distributed across the sphere. In contrast, for Nap, the forces are predominantly concentrated near the $xy$-plane.

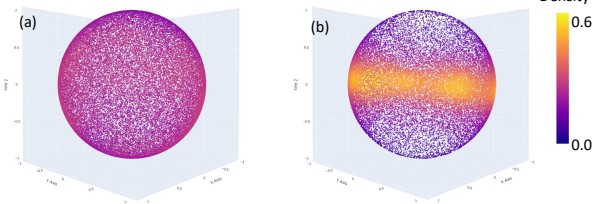

Figure 5: Distribution of force labels for (a) aspirin and (b) Nap in MD17. The force are uniformly distributed across the $x - z$ directions for aspirin, but predominantly distributed within the $xy$ plane for Nap.

ACKNOWLEDGMENTS

The authors acknowledge the use of artificial intelligence tools in the preparation of this manuscript. Specifically, the large language models Gemini, ChatGPT, and Claude were employed for three primary purposes: (i) to aid in improving the clarity and style of the text, (ii) for retrieval and discovery of related literature, and (iii) to explain certain concepts.

