# OpenReview forum: "On the Geometry of Uncertainty: Equivariant Uncertainty Estimation for Molecular Vector Properties"
_ICLR.cc/2026/Conference — Submitted to ICLR 2026_

### Official Review · Reviewer_ZU1A · 2025-10-31

**Soundness:** 3
**Presentation:** 1
**Contribution:** 3
**Rating:** 4
**Confidence:** 3

**Summary:**

The authors propose a new method for uncertainty estimation of vector properties (of molecules).
They construct a new prediction head to enable equivariant multivariate evidential regression. They provide equivariance of the covariance matrix by using exp(\alpha) I to the FF^T term. They evaluate the predicitve quality and calibration on several datasets including OOD splits.

**Strengths:**

- The idea seems simple yet effective
- The approach does not require sampling during inference
- Good empirical performance

**Weaknesses:**

- I would like to see a small hyperparameter study on the force loss contribution, as I believe it is quite critical to see how good the performance can be, also on energy, and on the tradeoff between energy and force prediction.
- The text can be improved: there are some typos, and for example, a missing reference to tables 3 and 4
- I would like to see more methods. The number of baselines seems a bit limited to me

**Questions:**

- In the results section, you introduce VBP of Liu, then in the table write VBD and in the text SBP. Are the first and the last typos and you mean VBD each time?
- Can you provide more insights and ablations on the effect of alpha?

---

> ### Author Response · Authors · 2025-11-26
>
> We thank you for taking the time to review our work.
>
> ---
>
> **W1. hyperparameter study on the force loss contribution**
>
> **R1:**  We appreciate the suggestion to analyze the sensitivity to the force loss coefficient. We have included this ablation study in Table 14 of the Appendix. The results indicate that a coefficient in the range $[0.99, 0.999]$ minimizes force MAE, while a range of $[0.8, 0.99]$ yields the lowest energy MAE. Notably, a very low force weight results in high errors for both metrics. This observation aligns with established findings from the Open Catalyst Project (OCP), where a high force loss weight is known to facilitate energy accuracy in Structure to Energy and Force (S2EF) tasks. (https://github.com/facebookresearch/fairchem/blob/3dd884682b289051579c0f9db79ea692c65f6195/src/fairchem/applications/AdsorbML/adsorbml/configs/painn.yml#L60) Overall, the results demonstrate that the fixed weight of 0.99 used in our study effectively balances energy and force accuracy.
>
> **W2: The text can be improved: there are some typos, and for example, a missing reference to tables 3 and 4**
>
> **R2:** We appreciate the reviewer pointing this out. We have corrected the missing references to Tables 3 and 4 (now Tables 4 and 5) and conducted a thorough proofreading of the manuscript to fix typos and improve clarity.
>
> **W3: I would like to see more methods. The number of baselines seems a bit limited to me**
>
> **R3:** We thank the reviewer for the suggestion. We have expanded our evaluation by adding results for CARD and MER. We also removed VBDDropout to ensure a comparison only between strong baselines.
>
>
> **Q1: In the results section, you introduce VBP of Liu, then in the table write VBD and in the text SBP. Are the first and the last typos and you mean VBD each time?**
>
> **RQ1:**  We thank the reviewer for identifying these inconsistencies. Yes, VBP and SBP are typos of VBD. As we have removed the VBD baseline in the revised manuscript, these acronym errors are no longer present.
>
> **Q2:  Can you provide more insights and ablations on the effect of alpha?**
>
> **RQ2:** We thank the reviewer for this insightful suggestion. Theoretically, as detailed in Section 3.2, $\alpha$ is crucial for ensuring numerical stability and solution uniqueness. Without this term, the factorization is invariant to right-multiplication by any orthogonal matrix $\mathbf{Q}$ (since $(\mathbf{F}\mathbf{Q})(\mathbf{F}\mathbf{Q})^\top = \mathbf{F}\mathbf{F}^\top$), which hinders optimization. This is confirmed empirically in Table 5, where the model fails to converge on the MD17 and QM7X datasets without $\alpha$. We further conducted an ablation study using only the scalar component (i.e., $\boldsymbol{\Psi} = \exp(\alpha)\mathbf{I}$). While this configuration achieves reasonable Mahalanobis distance-based calibration, it yields a strictly isotropic covariance matrix, failing to capture the directional uncertainties inherent in physical vector properties. Overall, both theoretical and empirical results support our covariance design, which uses $\alpha \mathbf{I}$ to ensure a unique solution and $\mathbf{F}\mathbf{F}^\top$ to provide directional uncertainties. We have added a discussion about this in lines 507-510, Page 10:
>
> *"... The configuration with $m=0$ forces the covariance matrix to be an isotropic diagonal matrix. While this yields good MCal (likely because the Mahalanobis distance can be minimized by a uniform uncertainty sphere), it inherently fails to capture directional uncertainties, limiting its physical utility. ..."*

---

### Official Review · Reviewer_36HM · 2025-11-01

**Soundness:** 1
**Presentation:** 3
**Contribution:** 2
**Rating:** 2
**Confidence:** 4

**Summary:**

This paper introduces Equivariant Multivariate Evidential Regression (EMER), a novel framework for uncertainty estimation (UE) of vector-valued molecular properties, such as atomic forces, which need to respect physical symmetries. This is achieved through a novel parametrization of the covariance matrix that is equivariant to rotations. A set of experiments on three molecular datasets using a newly introduced calibration metric is presented to compare the proposed method to some selected baselines.

**Strengths:**

* The problem of designing vector-valued uncertainty estimation methods that are equivariant to 3D rotations is interesting and impactful.
* The idea of extending Multivariate Evidential Regression (MER) by re-parametrizing the covariance matrix is sound.
* The paper is mostly clear and well-written.
* The authors do a good job in motivating their approach and discussing related works.

**Weaknesses:**

* The main issue with this paper is the experimental setup. Specifically, the authors propose a new calibration metric and rely solely on it to compare the different models and draw their conclusions. At the same time, this new metric is not sufficiently explained or analyzed, making it difficult to use as a reliable evaluation metric. The authors should (i) better explain this metric by comparing it to existing metrics (please add references in section 3.3) and analyzing it on known benchmarks, and (ii) use other metrics that are commonly used in the literature, such as the calibration score from [1].
* Another serious limitation is the following: the main motivation behind this work is that uncertainty estimation methods for vector-valued molecular properties should be equivariant to rotations; however, this need is neither experimentally validated nor reflected in the results. Specifically, the raw proposed method (EMER) consistently performs worse than Dropout. Given the absence of other baselines, it is challenging to assess the primary contribution of this paper (see the next point).
* The authors only compare to selected sampling-based methods. However, in order to assess the effect of adding equivariance to the UE method, the authors should compare it to related baselines that do not employ equivariance, such as the original MER [2] and LNK [1]. Specifically, the authors need to compare EMER with MER and EMER-Dropout with MER-Dropout, applying the same dropout approach to MER for a fair comparison.
* The presentation of the main contribution of this paper, which is the novel parametrization of the covariance matrix, is very ambiguous. Specifically, $F$ is sometimes referred to as a vector and at other times as a matrix. Also, it is not discussed how the proposed parametrization behaves in terms of over- or under-parametrization of the covariance matrix.

**References**

[1] Wollschläger, Tom, et al. "Uncertainty estimation for molecules: Desiderata and methods." International conference on machine learning. PMLR, 2023.

[2] Meinert, Nis, and Alexander Lavin. "Multivariate deep evidential regression." arXiv preprint arXiv:2104.06135 (2021).

**Questions:**

1. In Equation 2, what is $m$? Additionally, please formally define all variables, including their domains and dimensions, such as the NIW parameters and the Gamma function.
2. The current setup predicts a covariance matrix for each atom in the molecule. Can the method be extended to capture correlations between different forces and between forces and energy?
3. Is there any reference for the proposed calibration metric?
4. What are VBP, VBD, and SPD mentioned in the experiments section? What is "Benchmark" in Table 1?

---

> ### Author Response · Authors · 2025-11-26
>
> We sincerely appreciate you taking the time to review our work.
>
> ---
>
> **W1: the experimental setup**
>
> **R1:** We thank the reviewer for this constructive suggestion. We agree that benchmarking against community-standard metrics is vital for transparency and comparability. Per your request, we have expanded our evaluation in Table 1 (MD17), Table 3 (QM7-X) and Table 4 (OC20) to include both Raw Calibration Error (RCal) and Negative Log Likelihood (NLL).
> 1. As shown in the revised Table 1, while our proposed EMER-Dropout achieves the best Mahalanobis Calibration (MCal), the non-equivariant version (MER-Dropout, also ours) occasionally yields slightly lower (better) RCal and NLL scores on specific molecules (e.g., Naphthalene).
> 2. We investigated why a non-equivariant model would score well on dimension-wise calibration. We hypothesize that because standard datasets (like MD17) often provide pre-aligned structures, non-equivariant models overfit to the specific coordinate frame of the training set. This effectively "masks" their inability to handle geometric transformations.
> 3. To test this hypothesis—and demonstrate the necessity of our approach—we conducted a new Rotation Analysis (added as Table 2). We evaluated both models on the original test set and a rotated version.
>     * Non-Equivariant Baseline (MER-D): Performance collapses under rotation. For Naphthalene, RCal degrades catastrophically from $\sim0.12$ (static) to $\sim0.67$ (rotated).
>     * Ours (EMER-D): Remains robust. The calibration degradation is significantly lower, maintaining utility across orientations.
>
> Therefore, the results reinforce our claim: standard metrics on aligned data can give a false sense of reliability for vector quantities. MCal remains the superior metric for this domain as it is mathematically invariant to rotation and properly accounts for component correlations.
>
> **W2: equivariance requirement is not experimentally validated; improvement come from Dropout**
>
> **W3: compare EMER with MER and EMER-Dropout with MER-Dropout**
>
> **R2 & R3:** We appreciate the reviewer pushing for a rigorous isolation of the equivariance contribution. These two concerns go to the heart of our paper's motivation. To address them, we have:
>
> 1.	Added the requested baselines: We included MER and MER-Dropout in our evaluation (Table 1, 2, 4). Note: We excluded LNK as it is designed for scalar energy prediction via GPs and cannot natively being transferred to force.
>
> 2.	Experimentally Validated the Equivariance Requirement: On standard random splits, MER-Dropout and EMER-Dropout perform similarly. This is because standard test sets share the same directional bias as the training set (see Figure 4, newly added to Appendix). To rigorously test the motivation (W2), we evaluated both models on a rotated test set (Table 2). The results are decisive:
>
>     i.	MER-Dropout (Non-Equivariant): Fails to generalize. Its uncertainty estimates depend on the orientation, causing the Component RCal to spike catastrophically from $\sim0.15$ to $\sim0.70$ upon rotation.
>
>     ii.	EMER-Dropout (Equivariant): Remains robust, maintaining a calibration error of $\sim0.40$.
>
> This provides the missing experimental validation: non-equivariant uncertainty is physically unreliable because it cannot handle the geometric transformations inherent in molecular dynamics.
>
> 3.	Addressing "EMER < Dropout": The lower performance of raw EMER (and raw MER) is a known phenomenon in Deep Kernel/Evidential Learning called "feature collapse" ( (Ober et al., PMLR, 2021), where powerful encoders learn over-confident representations. This confirms that Dropout is a necessary regularizer for the evidential head, but—as the rotation experiment proves—it is insufficient for geometric correctness without our equivariant parameterization.
>
> **W4: Ambiguity of $\mathbf{F}$ and Parameterization Analysis**
>
> **R4:** We thank the reviewer. We have revised Section 3.2 to explicitly define $\mathbf{F} \in \mathbb{R}^{3 \times m}$ as a matrix, where $m$ controls the rank of the anisotropic component. We justify our choice of $m=2$ via a new ablation study (Appendix Tables 5 & 7) and SVD analysis (Figure 4):
>
> 1. Over-parameterization ($m=3$): As discussed in Section 3.2, a full-rank factorization creates non-unique solutions. Table 5 shows that this leads to training instability and failure to converge.
>
> 2. Under-parameterization ($m < 2$): Setting $m=0$ leads to a isotropic covariance matrix, while $m=1$ restricts uncertainty to a single principal direction. This fails to model planar correlations typical in molecules; for Nap, $m=1$ causes severe miscalibration along the orthogonal $z$-axis (RCal degrades to 0.487, Table 7).
>
> 3. Optimal Fit ($m=2$): This setting provides the necessary expressiveness for planar symmetries without instability. Our SVD analysis (Figure 4) confirms that the first two singular values account for $>90$% of the variance, validating a rank-2 approximation is physically sufficient.

---

> ### Author Response · Authors · 2025-11-26
>
> **Q1: formally define all variables**
>
> **RQ1:** We sincerely thank the reviwer for pointing this issue. $m$ means the NIW parameteris, to improve clarity, we have changed it into $L_{\text{NLL}_i} = -\log p(\mathbf{y}_i | \boldsymbol{\mathbf{u}}_0, \kappa, \boldsymbol{\Psi}, \nu)$. We have also formally define the domains and dimensions of NIW parameters  in lines 209-214, Page 4.
>
> **Q2: extended to different forces and between forces and energy?**
>
> **RQ2:** We appreciate this insightful question, as it touches on the fundamental physical coupling between atomic forces (related to the Hessian) and energy.
>
> 1. Correlations between different forces (inter-atomic): While theoretically possible, extending EMER to capture correlations between all pairs of atoms would require predicting a global $3N \times 3N$ covariance matrix. This raises the computational complexity from $O(N)$ (linear) to $O(N^2)$ or $O(N^3)$. We deliberately designed EMER as a local, per-atom method to maintain the scalability required for large systems like OC20.
>
> 2. Correlations between force and energy: This presents a unique geometric challenge. Energy is rotationally invariant (scalar), while forces are equivariant (vectors). Constructing a joint covariance matrix that respects these mixed symmetries requires a specialized block-diagonal parameterization (handling scalar-vector cross-correlations).
>
> While these extensions are beyond the scope of this work, EMER provides the critical foundational step: solving the equivariant covariance parameterization for the vector components. We view the extension to joint energy-force distributions as a promising direction for future research.
>
> **Q3: Is there any reference for the proposed calibration metric?**
>
> **RQ3:** We appreciate this request for theoretical grounding. The proposed metric is not an ad hoc invention, but a direct application of the Chi-squared test for multivariate normality, which is the standard statistical method for validating multivariate distributions. Mathematically, if a predicted distribution $p(\mathbf{y}) = \mathcal{N}(\boldsymbol{\mathbf{u}}, \boldsymbol{\Sigma})$ is calibrated, the squared error normalized by the covariance (the Mahalanobis distance) must follow a Chi-squared distribution:
> $$(\mathbf{y} - \boldsymbol{\mathbf{u}})^\top \boldsymbol{\Sigma}^{-1} (\mathbf{y} - \boldsymbol{\mathbf{u}}) \sim \chi^2_n$$
> We have updated Section 3.3 to cite Bishop (2006) for the theoretical derivation.
>
> **Q4: What are VBP, VBD, and SPD mentioned in the experiments section? What is "Benchmark" in Table 1?**
>
> **RQ4:** We apologize for the confusion. We have removed the VBD baseline due to its poor performance to focus the analysis on competitive methods; thus, those terms have been removed. We have also corrected the column header in Table 1 from "Benchmark" to "Backbone".

---

### Official Review · Reviewer_L25o · 2025-11-01

**Soundness:** 3
**Presentation:** 3
**Contribution:** 3
**Rating:** 6
**Confidence:** 4

**Summary:**

The paper focuses and addresses a critical gap in machine learning for physical systems: the lack of reliable geometrically consistent uncertainty estimation (UE) for vector-valued properties, such as atomic forces and dipole moments. The authors highlight that while scalar UE naturally satisfies equivariance, applying standard multivariate UE techniques to vectors often violates fundamental SO(3) rotational symmetries. To this end, they introduce EMER, a sampling-free framework that extends evidential regression to satisfy geometric constraints.

**Strengths:**

- **Well-Motivated**: The paper is well-written and structured. The motivation of the paper is very clear: while uncertainty estimation (UE) for scalar properties in scientific machine learning is mature, it fundamentally fails for vector-valued properties (like atomic forces) because standard multivariate formulations violate essential SO(3) equivariance constraints. The proposed solution EMER introduces a novel contribution through its parameterization of the Normal-Inverse-Wishart scale matrix as $$\Psi(\alpha,F)=\exp(\alpha)I+FF^T$$, which combines an invariant scalar isotropic term with an equivariant rank-one term to guarantee both positive definiteness and equivariance, effectly removing the limitations of prior MER approaches that relied on non-equivariant Cholesky factorizations.
- **Reasonable Metrics**: Additionally, the authors found the drawbacks of the existing calibration metrics and proposed a rotation-invariant Mahalanobis distance-based metric for calibration to address the lack of suitable evaluation tools for the specific domain.
- **Empirical Breadth**: The experimental validation is comprehensive, utilizing three established benchmarks (MD17, OC20, QM7-X) and testing across multiple SORA backbone architectures (PaiNN, NequIP, MACE).

**Weaknesses:**

- **Lack Visualization**: The authors can think of a way to add more visualizations for readers to better understand their motivation and solution.
- **Metric Design**: Although the metric design sounds good to me for vector-valued problems, it still needs to convince people why not use traditional calibration metrics. Like why do not we use traditional calibration metrics, e.g., ECE, PICP, NLL, etc., on different dimensions of the vectors and take an average? This seems to be able to estimating the calibration of vectors?
- **Lack Discussion**: From the results, it seems that dropout itself is not that useful, but EMER+dropout is super useful. Can the authors elaborate more about this phenomenon?

**Questions:**

See Weaknesses

---

> ### Author Response · Authors · 2025-11-26
>
> **W1: Lack Visualization**
>
> **R1:** We thank the reviewer for this suggestion. We have moved the illustration diagram (formerly in the Appendix) to Figure 1 in the main text. This diagram, in together with Figure 2 in the Appendix, now provides an high-level view of how the equivariant covariance matrix is parameterized and constructed within the network flow. We have also added a new Figure 4 in the Supplementary Material which shows the unbalanced direction distribution of force vectors for Nap and Uracil in the MD17 dataset.
>
> **W2: Metric Design**
>
> **R2:** We sincerely appreciate this suggestion. Following your recommendation, we have expanded our evaluation to include dimension-wise metrics: the Raw Calibration Error (RCal, as defined in [LNK, 2023]) and Negative Log Likelihood (NLL). We report the maximum error across dimensions ($x, y, z$) to capture the worst-case performance.These additional results (added to Table 1 and the new Table 2) revealed a critical insight that strongly reinforces the motivation for our work:
> 1. Component-wise metrics give a false sense of security for non-equivariant models. As shown in Table 2, in the standard test orientation ("w/o rotation"), the non-equivariant baseline (MER-Dropout) appears well-calibrated with an RCal of $\approx 0.12$. However, simply rotating the test set causes catastrophic degradation (RCal explodes to $\approx 0.67$). This confirms that non-equivariant methods fail to generalize across geometric transformations, a fatal flaw for physical simulations that standard scalar metrics mask.
> 2. Equivariance leads to robustness. In contrast, our proposed EMER-Dropout remains significantly more robust under rotation. While component-wise RCal degrades slightly due to the difficulty of aligning marginal distributions in arbitrary frames, our method avoids the massive failure mode seen in the baseline.
> 3. Why Mahalanobis (MCal) is necessary. The instability of RCal under rotation highlights why we proposed the Mahalanobis-based metric. Vector physical properties are geometric objects where components ($x, y, z$) are correlated. MCal is mathematically rotation invariant:$$d^2 = (\mathbf{v} - \boldsymbol{\mu})^\top \boldsymbol{\Sigma}^{-1} (\mathbf{v} - \boldsymbol{\mu})$$It consistently evaluates the quality of the uncertainty ellipsoid regardless of the coordinate frame.
>
> **W3: Lack Discussion**
>
> **R3:** We thank the reviewer for raising this concern. We have replied this in Reviewer VPCF’s W5. We have also rewrite the Avoiding pitfalls in the Section 3 to make this clear, as
>
> *“… This phenomenon has been observed previously in deep kernel learning and is referred to as feature collapse. (Ober et al., 2021; van Amersfoort et al., 2021) … In comparison to Monte Carlo Dropout (Gal & Ghahramani, 2016), where dropout is applied during both training and inference, here dropout is used only during the training stage. As a result, it serves solely as a regularization mechanism and does not provide UE. This regularization strategy has also been successfully employed in Wollschlager et al. (2023) and Tran et al. (2019).”*

---

> > ### Comment · Reviewer_L25o · 2025-11-26
> >
> > Thank you for your detailed reply. My main concern has been solved. I will increase my score accordingly.

---

### Official Review · Reviewer_VPCF · 2025-11-05

**Soundness:** 3
**Presentation:** 3
**Contribution:** 3
**Rating:** 6
**Confidence:** 4

**Summary:**

This paper studies uncertainty estimation for multivariate SO(3)-equivariant regressors (e.g., vector forces, dipoles). Prior UQ methods either break equivariance (single-pass, scalar-oriented approaches) or require expensive sampling like ensembles/MC dropout; when they do sample, they only indirectly yield equivariant covariances and are costly at inference. The authors propose EMER (Equivariant Multivariate Evidential Regression), a single-pass, sampling-free framework that parameterizes uncertainty so it is provably equivariant, positive-definite, and identifiable, and pairs it with a rotation-invariant calibration test based on Mahalanobis distance behavior for vectors. They also introduce a lightweight dropout-regularized variant to stabilize the encoder and keep predicted covariances well-conditioned. Across MD17, QM7-X, and OC20-style setups, EMER delivers state-of-the-art accuracy with markedly better calibration than ensembles, Bernoulli dropout, and variational Bayesian dropout, with ablations showing why the isotropic term and a low-rank design are crucial for stability.

**Strengths:**

1. Uncertainty estimation is crucial for scientific discovery, especially in molecular and physical systems where data naturally follow equivariance constraints. The paper is among the first to explicitly address uncertainty estimation in equivariant settings.

2. The proposed EMER framework is conceptually clear, mathematically grounded, and provides a provably equivariant and positive-definite uncertainty parameterization.

3. Experiments across multiple benchmarks (MD17, QM7-X, OC20-style) show strong performance both in predictive accuracy and uncertainty calibration.

**Weaknesses:**

1. While the paper emphasizes the inefficiency of sampling-based methods, it does not compare against more recent generative or diffusion-based uncertainty estimation approaches (e.g., Two for One: Diffusion Models and Force Fields for Coarse-Grained Molecular Dynamics).

2. The method’s efficiency comes from its parameterized uncertainty form, which may limit expressiveness compared to non-parametric sampling-based alternatives. A discussion or quantitative comparison of this trade-off would strengthen the claims.

3. The experimental section primarily compares to older dropout and variational Bayesian dropout methods. Including more contemporary or stronger ensemble baselines would make the empirical validation more convincing.

4. While calibration metrics are useful, demonstrating how the proposed uncertainty improves downstream tasks such as active learning, model-based control, or molecular discovery would highlight practical impact.

5. Table 1 suggests that most calibration improvements may stem from dropout regularization rather than the new covariance parameterization; a clearer disentanglement of these effects would be helpful.

**Questions:**

See the Weaknesses section.

---

> ### Author Response · Authors · 2025-11-26
>
> We sincerely thank the reviewer’s positive comment and suggestions. You suggest has guided us to reconsider our parameteried form of $\boldsymbol{\Psi} = \exp(\alpha) \mathbf{I} + \mathbf{F}\mathbf{F}^\top$, which we now conclude a dimension of 2 is necessary for $F$ to better reserve the dimensional uncertainty.
>
> **W1: more recent generative or diffusion-based uncertainty estimation approachs**
>
> **R1**: We appreciate the suggestion to include diffusion-based baselines and have added CARD (Han et al., NeurIPS 2022) to our revised experiments. Regarding the "Two for One" method (Arts et al., 2023), we respectfully note it is not a suitable baseline for this study. That framework is designed for coarse-grained MD using a force-free training paradigm on equilibrium positions, yielding a smooth potential of mean force. Conversely, our work targets all-atom force fields with stiff potential energy surfaces using force matching. We acknowledge, however, that a unified diffusion framework for generation, dynamics, and uncertainty represents a promising future direction.
>
> **W2: A discussion or quantitative comparison of  the parameterized uncertainty form**
>
> **R2**: We thank the reviewer for highlighting the critical trade-off between the efficiency of parameterized methods and the expressiveness of non-parametric approaches. We agree that while parameterized forms (like our NIW prior) restrict the uncertainty landscape to unimodal distributions, this constraint is often physically valid for atomic forces, which typically exhibit Gaussian-like noise around equilibrium states.
> However, to fully address your concern regarding expressiveness, we conducted an additional ablation study comparing the dimensionality ($m$) of our equivariant factor matrix $\mathbf{F}$ (Table 11 in the Appendix). We found that while a rank-1 parameterization ($m=1$) is efficient, it lacks the expressiveness to capture anisotropic uncertainty in planar molecules (e.g., Naphthalene, Uracil). In these cases, $m=1$ failed to calibrate the $z$-axis (out-of-plane) uncertainty independently from the $x-y$ plane.
> By increasing the rank to $m=2$, the model gained sufficient expressiveness to decouple these orthogonal uncertainty components, significantly improving component-wise calibration without incurring the computational cost of sampling. This demonstrates that a low-rank parameterized covariance strikes the optimal balance: it is expressive enough to model physical anisotropy but orders of magnitude faster than ensembles. We included the results in Table 11 (Appendix) and added a discussion in Section 4.1 (lines 513-519, Page 10).
>
> *"...To further investigate the trade-off between parameter efficiency and model expressiveness, we analyzed the performance difference between $m=1$ and $m=2$. While $m=1$ and $m=2$ yielded comparable performance on standard metrics, we evaluated their robustness on molecules with anisotropic distribution, specifically Nap and Uracil (Table 11). For them a rank-1 parameterization ($m=1$) lacked the expressiveness to decouple in-plane and out-of-plane uncertainties, resulting in poor RCal along the $z$-axis. In contrast, $m=2$ maintained robust calibration across all dimensions. Consequently, we adopted $m=2$...."*
>
> **W3: Including more contemporary or stronger ensemble.**
>
> **R3:** We appreciate this suggestion to strengthen our empirical validation. In response, we have expanded our experiments to include two contemporary, competitive baselines: (1) CARD (Han et al., NeurIPS 2022), a state-of-the-art diffusion-based uncertainty method, and (2) Multivariate Evidential Regression (MER) \citep{mer_2021}, the direct non-equivariant counterpart to our approach. The results (added to Table 1, Table 2, Table3, and Table 4) demonstrate two critical findings:
>
> 1.	Competitive Accuracy: EMER achieves state-of-the-art force accuracy, consistently outperforming CARD and EMR (lower force MAE in 5/7 molecules on MD17).
>
> 2.	The Necessity of Equivariance: The comparison with MER highlights the specific failure mode of non-equivariant methods. We evaluated both models on a "rotated test set" where force vectors are aligned away from the training distribution's dominant plane (Table 2). MER-Dropout (non-equivariant) fails to generalize, with Raw Calibration error failing from $\sim0.15$ to $\sim0.70$ under rotation. EMER-Dropout (equivariant) maintains robust calibration ($\sim0.40$), proving that equivariance is not just a theoretical constraint but a practical requirement for reliable uncertainty in physical systems.

---

> ### Author Response · Authors · 2025-11-26
>
> **W4: demonstrating how the proposed uncertainty improves downstream tasks**
>
> **R4:** We agree that demonstrating downstream utility is critical. To address this, we have added an active learning experiment on QM7X dipole moment prediction to the revised manuscript. Using a limited budget of 10 iterations (20 samples per step), we found that EMER and EMER-Dropout consistently achieved lower MAE compared to previous baselines (Dropout, MER), confirming that respecting geometric constraints improves sample efficiency (Figure 2 in the Appendix).
>
> Beyond active learning, EMER is uniquely suited for applications where geometric consistency is non-negotiable:
>
> 1. Adaptive QM/MM: In hybrid simulations, uncertainty triggers expensive QM calculations. Non-equivariant methods (e.g., MER) can artificially spike uncertainty due to rotation, triggering unnecessary computations. EMER ensures physically consistent decision boundaries.
>
> 2. Label Noise Detection: EMER provides a principled decomposition of aleatoric uncertainty, enabling the systematic detection of noisy force labels recently identified in molecular benchmarks (e.g., e.g., Kuryla et al., arXiv:2510.19774).
>
> **W5: a clearer disentanglement of the improvements by dropout and by the new covariance parameterization**
>
> **R5:** We thank the Reviewer for raising this concern. We clarify that dropout and the proposed equivariant parameterization serve distinct roles. We employ dropout solely during training to regularize the latent representation, which is an established practice in Deep Kernel Learning to prevent "feature collapse" (Ober et al., 2021; Wollschläger et al., 2023). This differs from Monte Carlo Dropout (Gal & Ghahramani, 2016), which is applied at inference for uncertainty estimation. Crucially, as shown in the updated Table 2, the equivariant parameterization is the primary driver of rotational robustness (EMER-Dropout vs MER-Dropout). Without it, the method's performance degrades significantly in rotated settings, even with dropout enabled. Thus, dropout stabilizes the latent space, while the new parameterization ensures the necessary equivariance.

---

### Author Response · Authors · 2025-12-03
**Summary of changes following rebuttal**

We would like to thank the Reviewers for their constructive suggestions, as well as the AC for the substantial work this year. We have addressed all the Weaknesses and Questions. We would also specifically like to thank Reviewer L25o for confirming that we resolved her/his concern and raising the score. We provide below a brief summary of our work and the revisions we made after the rebuttal.

**1. Summary of our contributions**

**(i)** As noted by Reviewers VPCF and L25o, our manuscript is among the first to explicitly address uncertainty estimation (UE) in equivariant settings. We formally formulated the problem of UE for equivariant vector outputs, addressing a gap in machine learning for physical systems.

**(ii)** We proposed the Equivariant Multivariate Evidential Regression (EMER) framework, which provides equivariant UE for vectors. Our results demonstrate that this method achieves state-of-the-art predictive accuracy and calibration.

**(iii)** We introduced a Mahalanobis distance-based calibration score for this problem, overcoming the limitation of standard scalar metrics in this setting.

**2. Modifications made according to Reviewers’ comments**

**(i) Experimental setting of metrics (Reviewer 36HM)**

Reviewer 36HM raised a concern that we initially only reported Mahalanobis distance–based calibration score. To address this, we added Raw Calibration Score and NLL for all main results (Tables 1, 3, and 4). Our method consistently outperforms baselines across both previous and new calibration metrices.

**(ii) Limited number of baseline methods (Reviewer VPCF, 36HM and ZU1A)**

Another concern was the limited number of baselines. To fully address this, we added the CARD baseline (suggested by Reviewer VPCF) and MER (suggested by Reviewer 36HM). Our proposed method continues to outperform these new baselines.

**(iii) Validation of the need for equivariant UE**

Reviewer 36HM also questioned whether equivariant UE is an experimentally validated requirement. To address this, we conducted new experiments in which we rotated molecules in the MD17 dataset (Table 2). Although the non-equivariant MER-Dropout performs comparably to EMER-Dropout on standard benchmarks, its good calibration is misleading because training and test sets share the same orientation distribution. When we simply rotated the test set, MER-Dropout failed dramatically, while EMER-Dropout remained robust. These results confirm that equivariant UE is a practical and necessary physical requirement.

**(iv) Other required experiments**

We also addressed all other requested experiments, including:
- a downstream active learning experiment (new Figure 2, Appendix), showing our equivariant method achieves the highest accuracy;
- an analysis of forcen loss weight (new Table 12, Appendix), confirming that our fixed weight of 0.99 is reasonable;
- an ablation study on the effect of $\alpha$ (updated Table 5 and Table 11), showing that $\alpha \mathbf{I}$ ensures a unique and stable solution, while $\mathbf{F}\mathbf{F}^\top$ captures directional uncertainty.


With these revisions, we have addressed all concerns and thank the Reviewers again for helping us improve our MS. We believe that the paper now meets the high standard of ICLR and appreciate the AC’s favorable consideration.

---

### Meta-Review · Area_Chair_wczB · 2026-01-07

**Summary:**

This paper proposes an equivariant uncertainty estimation framework for vector-valued molecular properties, introducing a new covariance parameterization and a Mahalanobis-distance–based calibration metric. Reviewers agreed that the problem of uncertainty estimation for equivariant vector outputs is important and that the proposed framework is technically motivated.

However, there were substantial concerns regarding the experimental design, evaluation methodology, and the strength of the empirical evidence supporting the paper’s main claims. In particular, the paper relies heavily on a newly introduced calibration metric without sufficiently validating or contextualizing it, and the experimental results do not convincingly demonstrate the necessity or benefit of equivariant uncertainty estimation in practice. While the authors provided a detailed rebuttal and additional experiments, these responses did not fully resolve the core concerns raised by multiple reviewers. Based on the overall review signal and remaining issues, the paper is recommended for rejection.

**Reviewer Concerns:**

After carefully reading the paper, the reviews, and the rebuttal, I largely agree with the key concerns raised by Reviewer 36HM. The most significant issue is the experimental setup and evaluation strategy. The paper introduces a new calibration metric and relies on it as the primary basis for comparing methods and drawing conclusions, but this metric is not sufficiently explained, validated against existing calibration metrics, or analyzed on well-understood benchmarks. Although additional metrics were added in the rebuttal, the paper’s main narrative and conclusions remain centered on the new metric, which weakens confidence in the evaluation.

A second major concern is that the core motivation for equivariant uncertainty estimation is not convincingly supported by the experimental results. In the original experiments, the proposed method without dropout underperforms standard dropout baselines, raising questions about whether equivariance itself is driving the observed improvements. While the rebuttal adds rotated-test experiments and additional baselines, reviewers remain unconvinced that these results provide a clear and general demonstration of practical advantage.

Finally, reviewers noted that the baseline comparisons are incomplete, particularly with respect to non-equivariant uncertainty estimation methods that are closely related to the proposed approach. Without systematic comparisons to these methods under matched settings, it is difficult to isolate the contribution of equivariance versus other modeling choices. Although the authors attempted to address these points in the rebuttal, the overall experimental evidence remains insufficiently convincing.

**Reviewer Scores:**

The reviewer scores were 6 (VPCF), 6 (L25o), 4 (ZU1A), and 2 (36HM).

Based on the discussion, Reviewer L25o explicitly indicated they would increase their score after the rebuttal. Even if Reviewer VPCF and Reviewer ZU1A had the opportunity to check the rebuttal, their scores would very likely have remained the same. While the rebuttal addressed several technical and presentation-related concerns, the additional clarifications and experiments in the rebuttal would be unlikely to materially change their overall assessment. Reviewer 36HM had strong and valid concerns and would be unlikely to increase their score even with further discussion.

---

### Decision · Program_Chairs · 2026-01-26

Reject